# The Marginal Ice Zone as a dominant source region of atmospheric mercury during central Arctic summertime

Fange Yue[1], Hélène Angot [2,3,4] ✉, Byron Blomquist[5,6], Julia Schmale [2], Clara J. M. Hoppe [7], Ruibo Lei[8], Matthew D. Shupe [5,6], Liyang Zhan[9], Jian Ren [10], Hailong Liu[11], Ivo Beck[2], Dean Howard[5,6], Tuija Jokinen [12,13], Tiia Laurila[12], Lauriane Quéléver[12], Matthew Boyer[12], Tuukka Petäjä [12], Stephen Archer [14], Ludovic Bariteau[5,6], Detlev Helmig[15], Jacques Hueber[16], Hans-Werner Jacobi [4], Kevin Posman[14] & Zhouqing Xie [1] ✉

Atmospheric gaseous elemental mercury (GEM) concentrations in the Arctic exhibit a clear summertime maximum, while the origin of this peak is still a matter of debate in the community. Based on summertime observations during the Multidisciplinary drifting Observatory for the Study of Arctic Climate (MOSAiC) expedition and a modeling approach, we further investigate the sources of atmospheric Hg in the central Arctic. Simulations with a generalized additive model (GAM) show that long-range transport of anthropogenic and terrestrial Hg from lower latitudes is a minor contribution (~2%), and more than 50% of the explained GEM variability is caused by oceanic evasion. A potential source contribution function (PSCF) analysis further shows that oceanic evasion is not significant throughout the ice-covered central Arctic Ocean but mainly occurs in the Marginal Ice Zone (MIZ) due to the specific environmental conditions in that region. Our results suggest that this regional process could be the leading contributor to the observed summertime GEM maximum. In the context of rapid Arctic warming and the observed increase in width of the MIZ, oceanic Hg evasion may become more significant and strengthen the role of the central Arctic Ocean as a summertime source of atmospheric Hg.

Mercury (Hg) is one of the most toxic heavy metals, and its long-distance transport in the air makes it a global pollutant[1]. Hg in the atmosphere exists in three forms based on its physical and chemical properties: gaseous elemental mercury (GEM or Hg(0)), and gaseous or particulate divalent Hg (Hg(II) and Hg(p))[2,3]. GEM has a long residence time in the air (0.5–1 year) due to its high volatility, low water solubility and low chemical reactivity and accounts for more than 90% of the total content of Hg in the surface air[4–6]. Gaseous and particulate divalent Hg, which can be formed by GEM oxidation, have a shorter lifetime (hours to weeks) and can be effectively eliminated from the air via dry and wet deposition[2,4,7].

The Arctic is an important component of the Northern Hemisphere Hg cycle and a sensitive area for environmental Hg exposure[8]. The latest Arctic Mercury Assessment Report (AMAP, 2021) and refs. 9,10 show that people living near the Arctic Circle have some of the highest Hg levels in the world due to their traditional diet. In addition, relatively long Arctic food chains can also result in Arctic wildlife being at high risk of Hg exposure. Long-range transport of air, riverine input and coastal erosion from circumpolar land can be primary Hg sources to the Arctic[11,12]. A fraction of these historical primary emissions can deposit and accumulate in seawater, plants, glaciers and ice sheets in the Arctic and later be re-emitted as secondary emissions

of atmospheric Hg[13–15]. Atmospheric Hg transport, transformation, and deposition pathways thus play important roles in the Arctic Hg cycle[10]. Since Schroeder et al.[16] first reported the occurrence of springtime atmospheric Hg depletion events (AMDEs) in Alert (Nunavut, Canada), a phenomenon during which GEM is near-quantitatively oxidized to Hg(II) leading to Hg deposition onto snow/sea-ice, many long-term observation studies on atmospheric Hg have been conducted at several Arctic coastal monitoring stations (e.g., Alert, Zeppelin in Svalbard, and Villum Research Station in Northern Greenland)[17–20]. Thanks to multi-year observations at these coastal monitoring stations, the unique seasonality of atmospheric Hg in the Arctic has been revealed. It is characterized by a springtime minimum, driven by AMDEs, followed by a summertime GEM maximum when mean concentrations typically exceed northern hemispheric background levels. However, the origin of the Arctic summertime GEM maximum is still a matter of debate in the community. A modeling study by ref. [21] suggested that long-range transport of Asian air is the most important primary source of atmospheric Hg to the Arctic in all seasons. Sommar et al.[22] and Fisher et al.[12] later attributed enhanced summertime GEM concentrations to oceanic Hg evasion in the Arctic Ocean fed by terrestrial Hg inputs (rivers and coastal erosion) based on observations and several sensitivity runs using the GEOS-Chem model. Recent isotopic work suggests, however, the dominant role of re-emissions from the Arctic cryosphere, while the role of terrestrial Hg from rivers and coastal erosion was found to be minor[23]. Hg re-emissions from the cryosphere could be fueled by AMDEs deposited Hg that remains in the snowpack until the melt season. That fraction of Hg retained in the snowpack from spring to summer is, however, still highly uncertain. Using stable isotopes, Douglas and Blum[13] found that 76 to 91% of AMDEs deposited Hg is re-emitted prior to snowmelt in Alaskan snow. However, Zheng et al.[14] estimated generally less photoreduction loss (0 to -60%) from snow at Alert, with an average of $20 \pm 31\%$. These differences may be due to the complex factors that can affect the amount of Hg emitted from the snowpack/sea-ice, such as solar radiation, Hg speciation in snowpack/sea-ice, halide and particulate matter concentrations in the snowpack, snow temperature, snowfall frequency, and upward latent heat flux[24–26].

Our current understanding of the Arctic Hg cycle mostly stems from coastal observations due to the lack of data over the central Arctic Ocean. This may limit our understanding of the factors controlling the atmospheric Hg cycle over the whole Arctic Ocean. For example, regional differences are expected for se-ice physical properties (e.g., dynamic deformation, thermal processes (melting and freezing)), and chemical composition. These factors are expected to have non-negligible impacts on key processes of the Hg cycle, such as redox reactions and snow re-emission of Hg, ultimately affecting air–sea exchange and atmospheric Hg concentrations in the central Arctic[27–29]. Therefore, relying only on long-term observations from coastal stations might provide an incomplete understanding of the atmospheric Hg cycle in the central Arctic and lead to uncertainties in the modeling of atmospheric Hg in that region[8].

This paper presents an analysis of summertime observations (June 9–September 30, 2020) of GEM in the central Arctic Ocean during the MOSAiC (Multidisciplinary drifting Observatory for the Study of Arctic Climate) international expedition. We focus here on (1) GEM concentrations and their variability in the central Arctic Ocean with a comparison to observations at Arctic coastal stations and (2) the identification of sources contributing to GEM levels in the central Arctic Ocean.

## Results and discussion
### Characteristics of GEM concentrations during the whole summertime in the Central Arctic Ocean
The concentration of GEM during the whole observation period (June–September) in the central Arctic Ocean ranged from 1.02 to 2.99 ng/m³, with an average (±standard deviation) of $1.54 \pm 0.27$ ng/m³ ($n = 31{,}176$; number of 5-min datapoints) (see Fig. 1). The GEM average from June to August ($1.60 \pm 0.28$ ng/m³, $n = 22809$; latitude range: 78.34°N to 90°N) is very similar to the observations at several nearby coastal Arctic stations during the same months in 2011–2015 (e.g., $1.63 \pm 0.37$ ng/m³ at Alert, $1.60 \pm 0.23$ ng/m³ at Zeppelin, and $1.63 \pm 0.37$ ng/m³ at Villum; $t$-test: $p > 0.01$)[17], indicating that the summertime GEM observations at these Arctic stations are comparable to the GEM levels in the central Arctic Ocean. In addition, our observations ($1.32 \pm 0.07$ ng/m³, $n = 3332$) are comparable to the observed GEM average ($1.40 \pm 0.61$ ng/m³; $t$-test: $p < 0.01$) during the same period (August 25–September 4) in 2012 in the central Arctic Ocean (80°N–87.6°N, 9.2°E–168.9°W), while with apparently lower standard deviation than the latter[28]. This difference in standard deviation may to some extent be attributed to the significantly higher and less variable sea-ice fraction ($0.92 \pm 0.07$) in this study, which inhibited the oceanic evasion of Hg and led to the apparently more stable variation of GEM concentration, compared with ref. [15] (with average sea-ice fraction of $0.36 \pm 0.32$).

### Monthly variations
Monthly averaged GEM concentrations reached a maximum in July: July ($1.80 \pm 0.32$ ng/m³, $n = 8264$) > June ($1.59 \pm 0.17$ ng/m³, $n = 5908$) > August ($1.42 \pm 0.13$ ng/m³, $n = 8639$) > September ($1.35 \pm 0.097$ ng/m³, $n = 8368$) (Fig. 2a). A similar feature is typically observed at coastal Arctic monitoring stations, such as Alert, Villum, and Zeppelin[17,19], implying that summertime observations at high-latitude Arctic coastal stations can generally capture the GEM monthly variation pattern in the central Arctic Ocean[22]. In addition, the time series of GEM concentrations (Fig. 1b) shows that GEM variability was the largest in July, with the highest hourly averaged GEM concentration (2.99 ng/m³). The highest GEM concentration was accompanied by high solar radiation (RD), but low carbon monoxide (CO) and sulfur dioxide (SO₂) levels, both typical proxies for anthropogenic emissions (Supplementary Fig. 1), indicating that this GEM peak was not associated with anthropogenic emissions but likely attributable to natural processes (e.g., photoreduction and oceanic re-emission). A similar low correlation between GEM and CO was also observed during previous summertime cruises in the Arctic[22] and Antarctic Ocean[23]. Previous studies suggested that the increased atmospheric GEM level during summertime in the Arctic could be associated with the re-emission of Hg from the snowpack and ocean, facilitated by higher temperature and solar radiation and the melting of sea-ice[15,17,18,23,30]. After August, the GEM concentrations gradually decreased and leveled off. In this study, we also found that the monthly averaged concentration of GEM has no consistency with CO (e.g., the monthly averaged CO concentration is the lowest during the summer, which is opposite to GEM, Fig. 2b). This seems to suggest a minor role of anthropogenic emissions on the GEM monthly variability during the summertime in the Arctic Ocean. This will be further discussed in Section 2.3.1.

Previous studies have observed AMDEs at Arctic coastal monitoring stations in June[18], but this phenomenon was not observed between June and September during the MOSAiC expedition. One potential reason could be the air temperature. Low air temperature favors the production of reactive bromine radicals[31,32] and the oxidation of GEM (by stabilizing the Hg(I) intermediate)[33] and therefore favors the occurrence of AMDEs. AMDEs were actually observed in June at Zeppelin station when the temperature was between −5 and −10 °C, which were the lowest temperatures recorded for that month[18]. The warmer air temperature (average: $-0.86 \pm 1.12$ °C) in June during the MOSAiC expedition likely explains why no AMDEs were observed at this time. It should, however, be noted that AMDEs were ubiquitous during the MOSAiC springtime (March–May) as reported in ref. [34].

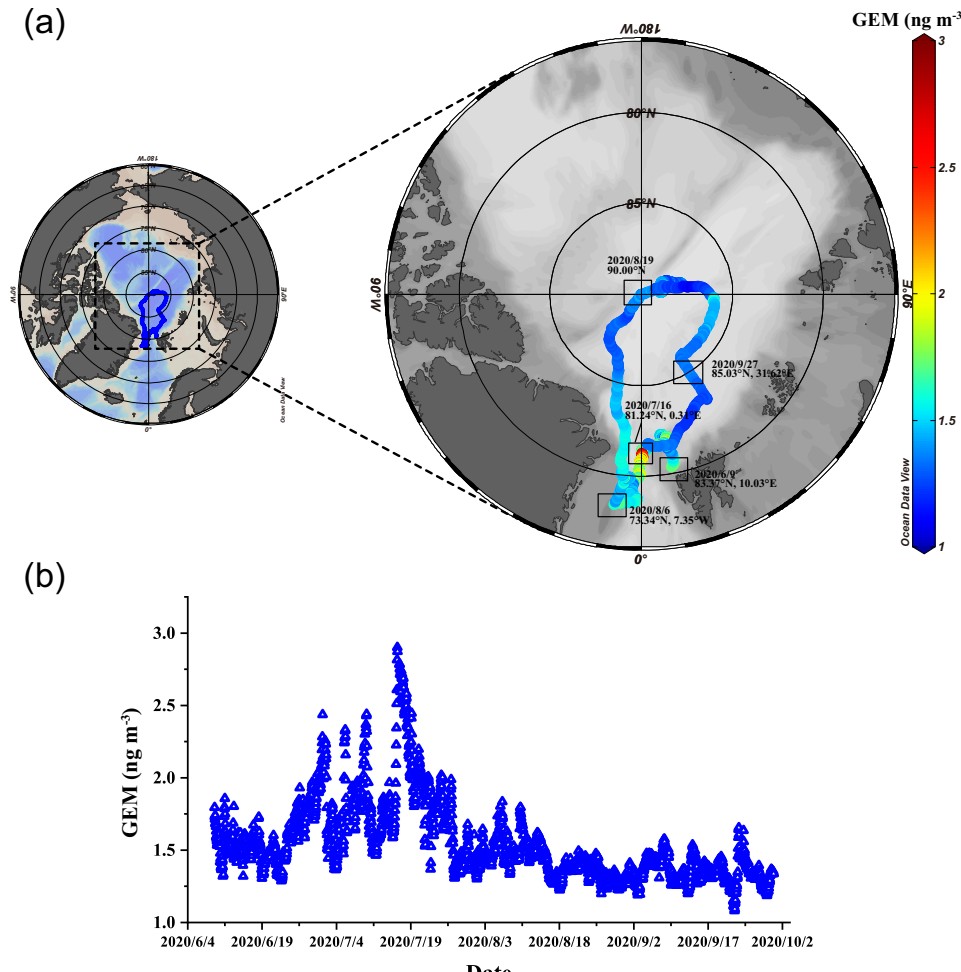

**Fig. 1 | Spatial distribution and time series of observed gaseous elemental mercury (GEM) concentrations in the research area of this study. a** Spatial distribution of observed gaseous elemental mercury (GEM) concentrations in the Arctic Ocean during the summer legs (June–September) of the Multidisciplinary drifting Observatory for the Study of Arctic Climate (MOSAiC) expedition. The color scale gives the GEM concentration. The figure was generated using Ocean Data View (https://odv.awi.de/)[85]. **b** The time series of GEM concentrations during the summer legs (June–September) of the MOSAiC expedition.

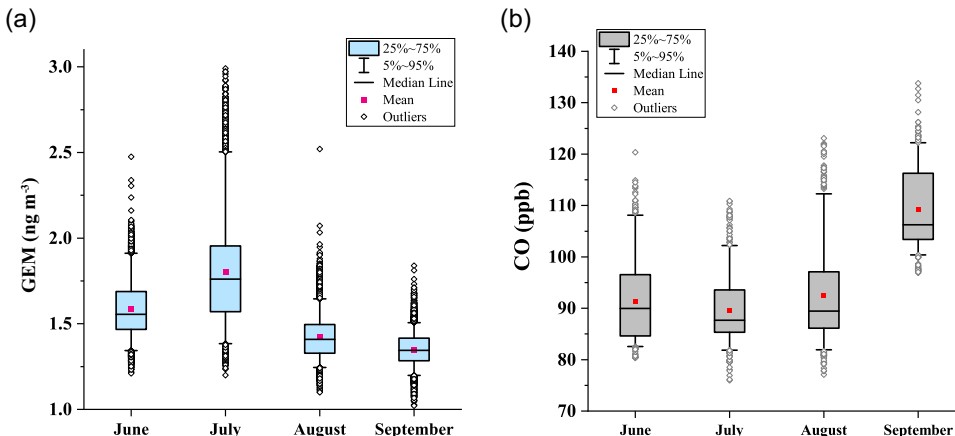

**Fig. 2 | Statistical information of gaseous elemental mercury (GEM) and carbon monoxide (CO) concentrations in each month.** Box plots of (**a**) gaseous elemental mercury (GEM) concentrations and (**b**) CO mixing ratios in June, July, August, and September.

## Sources of GEM in the central Arctic in summer

The summertime GEM maximum that follows springtime depletions (AMDEs) is a unique Arctic feature. This Arctic GEM seasonality suggests a transition from a "mercury sink" in spring to a "mercury source" in summer[24]. Thus, understanding the origins and corresponding mechanisms of this GEM variability is of great importance for accurately assessing the regional mercury budget and associated ecological consequences. However, the origin of this summertime GEM peak is

**Table 1 | Relative importance of each predictor including carbon monoxide (CO), the distance of the endpoint of each 48 h trajectory from the corresponding cruise observation location (Traj48h), the open-water fraction at the Polarstern position (Open-water), wind speed (WS), surface air pressure (P) and air temperature (Temp), and their corresponding identifications and contributions**

| Predictors | p value | Contribution | Identifications |
|---|---|---|---|
| CO | 4.78e-8 | 2% | Anthropogenic emissions |
| Traj48h | <2e-16 | 9% | Regional transport |
| Open-water | <2e-16 | 52% | Local oceanic emissions |
| P | <2e-16 | 9% | Meteorological conditions |
| WS | <2e-16 | 8% | |
| Temp | <2e-16 | 20% | |

still not entirely understood. To further understand the sources of GEM in the central Arctic Ocean summertime, we used a GAM (generalized additive model) simulation to evaluate the relative contributions of (1) the long-range transport of anthropogenic emissions, (2) local oceanic emissions and (3) meteorological conditions. Since the F test results from GAM imply the variance contribution of each predictor to the response, the relative importance of each predictor can be determined by the result of the F value for each predictor divided by the sum of all F values[35,36]. The results are displayed in Table 1 and discussed below.

The GAM results show that anthropogenic emissions associated with CO contribute only ~2% of the GEM variation. Its partial response curve shows that with increasing CO, GEM does not increase (Fig. 3a). Furthermore, the GAM partial response curve of Traj48h, which represents the regional transport potential of GEM, indicates that a positive relationship between GEM and Traj48h occurs in the low Traj48h range (<500 km). However, as Traj48h continues to increase, GEM generally decreases or remains at the same level (Fig. 3b). As a result, the GAM simulations indicate that the long-range transport of GEM is less significant than the regional transport over the ocean (<500 km from the ship). This hypothesis is further supported by the statistical analysis of air-mass back trajectories: the fraction of the 168 h backward trajectory's transport time in land area over the total transport time (Land fraction), which characterizes the potential contribution of land-based emissions to regional transport, was calculated every 2 h during the observation period. We find that the average GEM concentration slightly decreases as the land fraction increases (Fig. 3c), which to some extent suggests a minor contribution of land-based Hg emissions to GEM in the central Arctic Ocean during summertime. One potential explanation for the minor contribution of land-based Hg emissions to summertime GEM levels in the central Arctic is the position of the Arctic dome. The Arctic dome acts as a transport barrier for air masses and isolates the Arctic lower troposphere from lower latitudes, driven by thermal stratification of the lower atmosphere within the polar dome[37]. The Arctic dome can extend to approximately 40°N in wintertime, therefore favoring the transport of mid-latitude (i.e., North of 40°N) Hg emissions, and recede poleward to roughly north of 70°N in summertime, isolating the central Arctic from mid-latitudes[38,39]. More information on the seasonality of poleward pollution transport from mid-latitudes can be found in Bozem et al.[40] or Boyer et al.[41]. This is also reflected in the 7-day backward trajectories, which show that air masses were mainly concentrated over the central Arctic Ocean during the observation period (Supplementary Fig. 2). Moreover, the most recent AMAP/UNEP global Hg emission inventory shows few anthropogenic Hg sources in land areas around the Arctic Ocean[11], which supports the low contribution of anthropogenic land-based Hg emissions to the central Arctic Ocean in this analysis. Similarly, Skov et al.[19] reported a low contribution of anthropogenic

emissions (14–17%) to the GEM concentrations at the Villum station in northern Greenland.

The GAM analysis indicates that the open-water fraction (proxy for oceanic emissions) accounts for ~52% of the GEM variation during the observation period, highlighting the dominant role of oceanic emissions. This result is further supported by the statistical analysis of air-mass back trajectories: the fraction of the 168 h backward trajectory's transport time in marine area over the total transport time (Sea fraction (Traj)) was larger than 0.9 during most of the observation period, including the periods when GEM peaked (Fig. 4a). A previous study based on GEM data collected at Zeppelin station combined with the Lagrangian particle dispersion model FLEXPART also found that the highest GEM concentrations in July and August were associated with air masses from the marine boundary layer, suggesting the likely contribution of oceanic Hg evasion[15]. It is generally assumed that the presence of sea-ice inhibits the release of GEM from the ocean to the air[42]. Here, the partial response curve of the open-water fraction shows two obvious GEM peaks associated with low open-water fractions (approximately 0.2 and 0.5, respectively, Fig. 4b), which may be due to transport from regions with a higher open-water fraction.

To further identify GEM source regions, we conducted a PSCF (potential source contribution function) analysis (Fig. 5a), with 2,244 trajectories for hourly averaged GEM data. The PSCF results combined with the spatial distribution of sea-ice (Fig. 5a, b and Supplementary Fig. 3) show a GEM source region in the low-latitude Marginal Ice Zone (MIZ) between Greenland and Svalbard (black box area in Fig. 5a, b), in line with the hypothesis of a regional transport contribution. Our results, therefore, suggest that oceanic Hg evasion is not significant everywhere in the central Arctic Ocean but mainly occurs in the MIZ. This is in line with previous studies highlighting low dimethylsulfide (DMS) levels in the central Arctic and concentration spikes due to regional transport from the MIZ[43–45] (Fig. 5c, d). In addition, multi-year observations of GEM at several Arctic coastal stations showed that the summertime GEM peak is relatively weaker at Zeppelin station than at Alert and Villum stations, likely due to the altitude (474 m a.s.l.) and relative remoteness of that station from the MIZ[17]. These observations support the hypothesis that the MIZ is an important source of GEM in summer. Below, we discuss three hypotheses for this oceanic Hg evasion hotspot in the MIZ:

(1)   The load of Hg(II) substratum in the surface seawater, which determines the upper limit of Hg that can evade into the atmosphere;

(2)   The reduction capacity of Hg(II) in the surface seawater, which determines how much GEM (or dissolved gaseous mercury; the main evading form of Hg) is available;

(3)   The physical environment of the surface sea layer (e.g., presence of sea-ice, meltwater, and seawater) that influences air–sea gas exchange.

Firstly, the sea-ice-covered central Arctic Ocean is the location where springtime AMDEs occur; therefore, it receives and stores large amounts of Hg(II)[8,33,46]. This is supported by springtime GEM observations (March–May) during this expedition, which showed that AMDEs were ubiquitous over the central Arctic[34]. According to a modeling study, only 4% of deposited Hg(II) over the central Arctic Ocean is re-emitted before snowmelt due to the presence of halides that inhibit Hg(II) photoreduction[34]. In summer, this previously deposited Hg(II) is transferred to the surface ocean with snow and sea-ice melt[47], resulting in a large Hg load in surface seawater of the MIZ. This hypothesis is supported by the latest Arctic Hg mass balance budget by ref. 8: approximately 9 tons (4 ~ 15 tons) of Hg are stored in the Arctic Ocean sea-ice with sea-ice melt contributing 1.4 ± 0.4 tons of Hg to the Arctic Ocean.

Secondly, we observed a significant correlation between the daily GEM concentrations in ambient air and chlorophyll-a (Chla) concentrations in surface seawater, characterized by similar variation

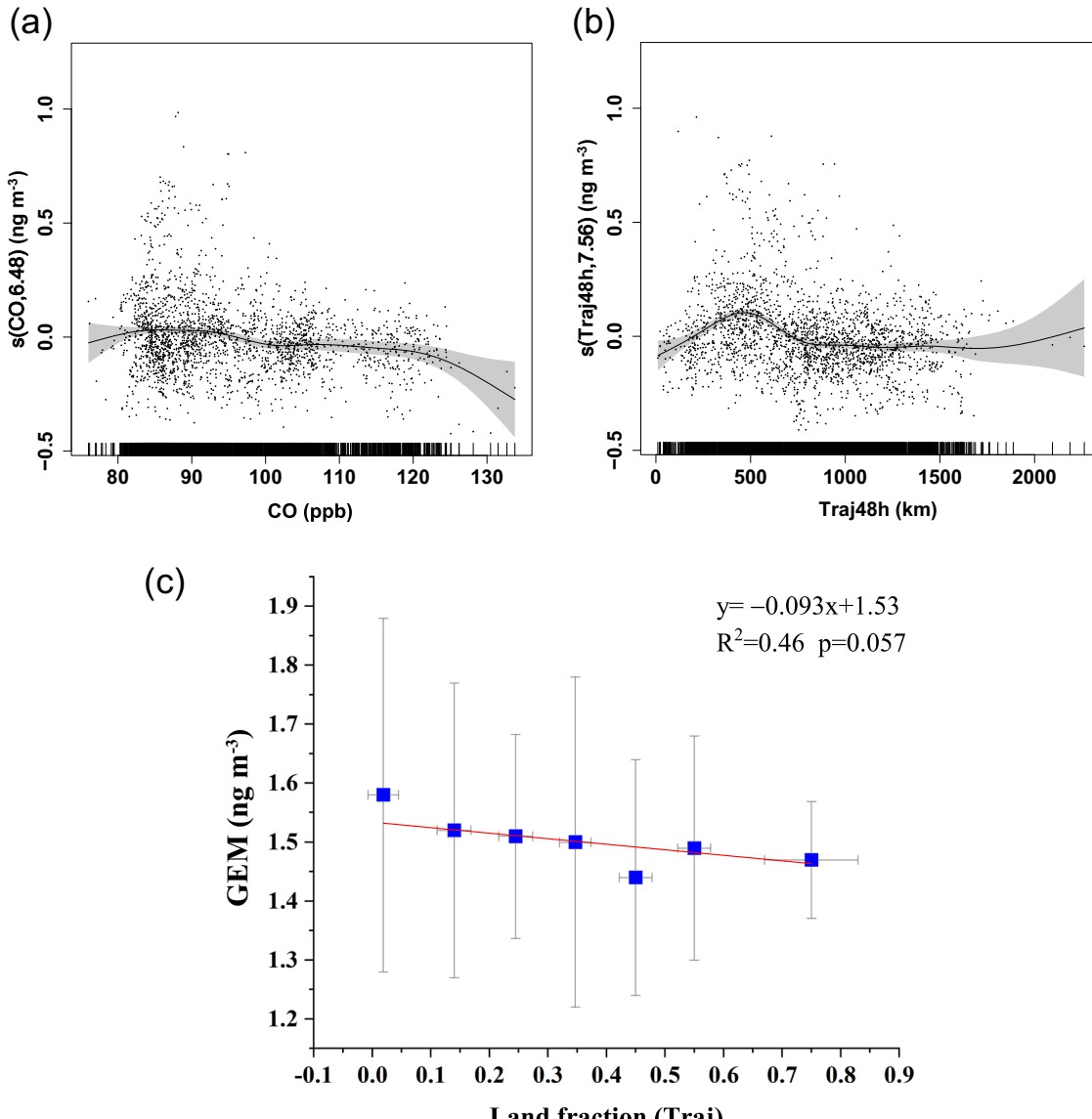

**Fig. 3 | Spline and relationship of gaseous elemental mercury (GEM) to carbon monoxide (CO) and the distance of the endpoint of each 48h trajectory from the corresponding cruise observation location (Traj48h).** Spline of gaseous elemental mercury (GEM) to (**a**) CO and (**b**) Traj48h (the distance of the endpoint of each 48 h trajectory (hourly resolution) from the corresponding cruise observation location). The y-axis in each subplot represents the smooth function term of each predictor with the estimated degrees of freedom inside the brackets. The gray area around the line is the 95% confidence bound for the response. **c** Relationship between the fraction of the 168 h backward trajectory's transport time over land area over the total transport time (Land fraction (Traj)) and its corresponding average GEM concentration. The error bars of x-axis and y-axis indicate the standard deviations of the average Land fraction (Traj) and the corresponding average GEM concentrations, respectively.

trends during the whole summer period, and a significant linear correlation between GEM and Chla ($R^2 = 0.43$, $P < 0.001$, Fig. 6). The Chla measurements in July (when Polarstern was located in the MIZ) indicate that a phytoplankton bloom occurred in the MIZ during the sampling period. This is a common feature of the MIZ due to increased stratification and reduced light limitation caused by melting snow and ice under conditions when surface nutrients are not yet limiting[48,49]. Several previous laboratory studies have suggested that the presence of marine phytoplankton can facilitate Hg(II) reduction to dissolved gaseous Hg[3,50,51]. The observed significant positive correlation between GEM and Chla could support the hypothesis that biological activity, which is particularly high in the MIZ in summer, facilitates the production of GEM (or Hg(0)). Different potential mechanisms have been suggested in the past: (1) the excretion of photoreactive organic compounds by phytoplankton can facilitate the photoreduction of

Hg(II) to Hg(0) via electron transfer[3,50,51]. This hypothesis may be supported by the occurrence of concurrent high solar radiation and high levels of particulate organic carbon (POC) observed in the surface seawater during the GEM and Chla peaks (Fig. 6 and Supplementary Fig. 4). (2) Phytoplankton can directly contribute to the reduction of Hg(II) to GEM through an enzymatic detoxification mechanism linked to their photosynthetic activity[3,52]. These mechanisms could explain why the MIZ has a high Hg(II) reduction capacity.

Thirdly, the melting of sea-ice in the MIZ removes the physical barrier to air–sea gas exchange[40], facilitating the re-emission of GEM. Contrary to the MIZ where the action of waves efficiently mixes the upper ocean, a clear vertical stratification of the upper ocean and the presence of a meltwater layer (10 cm to 1 m; fed by the rapid melt of snow and sea-ice during summer) was observed throughout summer in the central Arctic[53,54]. Previous studies and data collected during

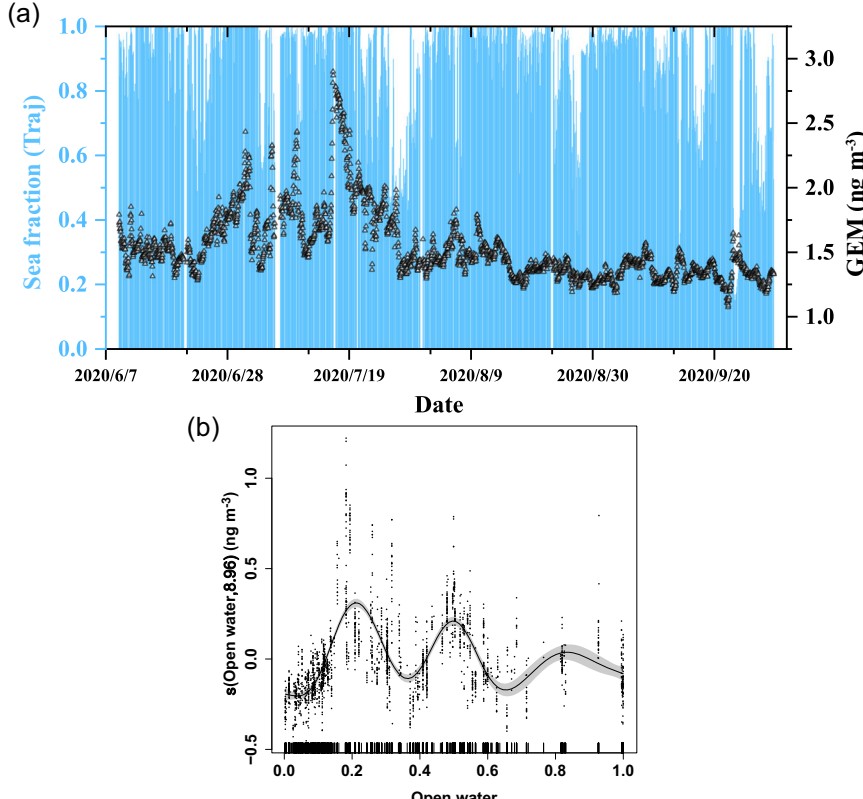

**Fig. 4 | Time series of Sea fraction (Traj) and spline of gaseous elemental mercury (GEM) to open water fraction (open water). a** Time series of the fraction of the 168 h backward trajectory's transport time over sea area over the total transport time (Sea fraction (Traj)) and GEM concentration. **b** Spline of gaseous elemental mercury (GEM) to open water fraction (open water).

MOSAiC suggest that the presence of this meltwater layer alters exchanges across the air–sea interface and limits water-to-air fluxes of key trace gases[54]. This vertical stratification along with a less biologically active central Arctic (i.e., lower reduction capacity; point (2)) could explain why ocean Hg evasion is limited north of the MIZ. GEM measurements performed during a poleward transect in early August during the MOSAiC expedition support this hypothesis, as GEM concentrations obviously decreased when *Polarstern* entered the pack ice with increased distance from the MIZ (Supplementary Fig. 5).

Overall, we suggest that oceanic evasion in the MIZ combined with regional transport is the dominant GEM source in the central Arctic—a phenomenon that likely explains the observed summertime GEM maximum. The main processes that drive significant oceanic Hg evasion in the Arctic MIZ are summarized in Fig. 7. A back-of-the-envelope calculation (see Supplementary Discussion) suggests a Hg evasion flux of 56 ng·m$^{-2}$·day$^{-1}$ in the MIZ. This flux is more than twice the flux measured in the Arctic open ocean (<24 ng·m$^{-2}$·day$^{-1}$)[42,55], but lower than that measured in coastal areas of the Canadian Arctic Archipelago (-130 ng·m$^{-2}$·day$^{-1}$)[56]. It is also an order of magnitude higher than the Arctic Ocean evasion flux reported by ref. 8 in their latest Arctic Hg mass balance budget (23–45 tons/year, i.e., 3.7–7.3 ng·m$^{-2}$·day$^{-1}$ assuming a surface area of $1.7 \times 10^7$ km$^2$). Finally, this flux is also higher than the evasion flux over continental shelf areas driven by riverine inputs[57]. Further dedicated field and modeling studies are needed to further constrain that flux.

**Impact of meteorological factors**
Meteorological conditions (wind speed, atmospheric pressure and air temperature) can influence the re-emission, transport and dilution of atmospheric Hg. Here, we find that the contribution of meteorological predictors to GEM variation in the central Arctic Ocean is up to 37%.

Among these, air temperature (Temp) is the major contributor (20%). The partial response curve of Temp shows an obvious upward trend of GEM in the Temp range of 260 - 275 K (Supplementary Fig. 6b). Higher temperature could not only promote the release of Hg from the surface ocean and snowpack[17], but also inhibit the oxidation process of GEM by reactive halogen radicals through causing the thermal decomposition of Hg(I) intermediate[33]. Alternatively, this partial response curve may reflect the transport of warmer air masses from the MIZ, which is supported by the PSCF analysis of Temp: for Temp in the range of 260 - 275 K, PSCF values are highest for the MIZ (Supplementary Fig. 7a). However, for Temp higher than 275 K, the contribution from the MIZ decreases significantly, which corresponds to the decreasing trend of PSCF values when Temp is larger than 275 K (Supplementary Fig. 7b). These results indirectly support the important role of oceanic evasion and regional transport of Hg from the MIZ in the central Arctic.

Atmospheric pressure (P) contributes 9% to GEM variation. The partial response curve of GEM and P generally shows a negative relationship (Supplementary Fig. 6c). This may to some extent be due to the fact that that lower surface air pressure conditions tend to produce low-level convergence, which would gather the GEM through the low-level compensated air mass (e.g., Ekman pump) and result in a GEM increase (e.g., the highest hourly GEM in this study was observed near a low-pressure center, Supplementary Fig. 8)[58,59].

Wind speed (WS) contributes 8% to GAM variance. In stagnant meteorological conditions characterized by a low WS range (e.g., <2.5 m/s in this study), the (re)-emitted Hg from the ocean might tend to accumulate, causing the positive correlation between GEM and WS[36]. As WS increases, this may enhance the atmospheric dilution of GEM, leading to the decreasing trend (Supplementary Fig. 6a).

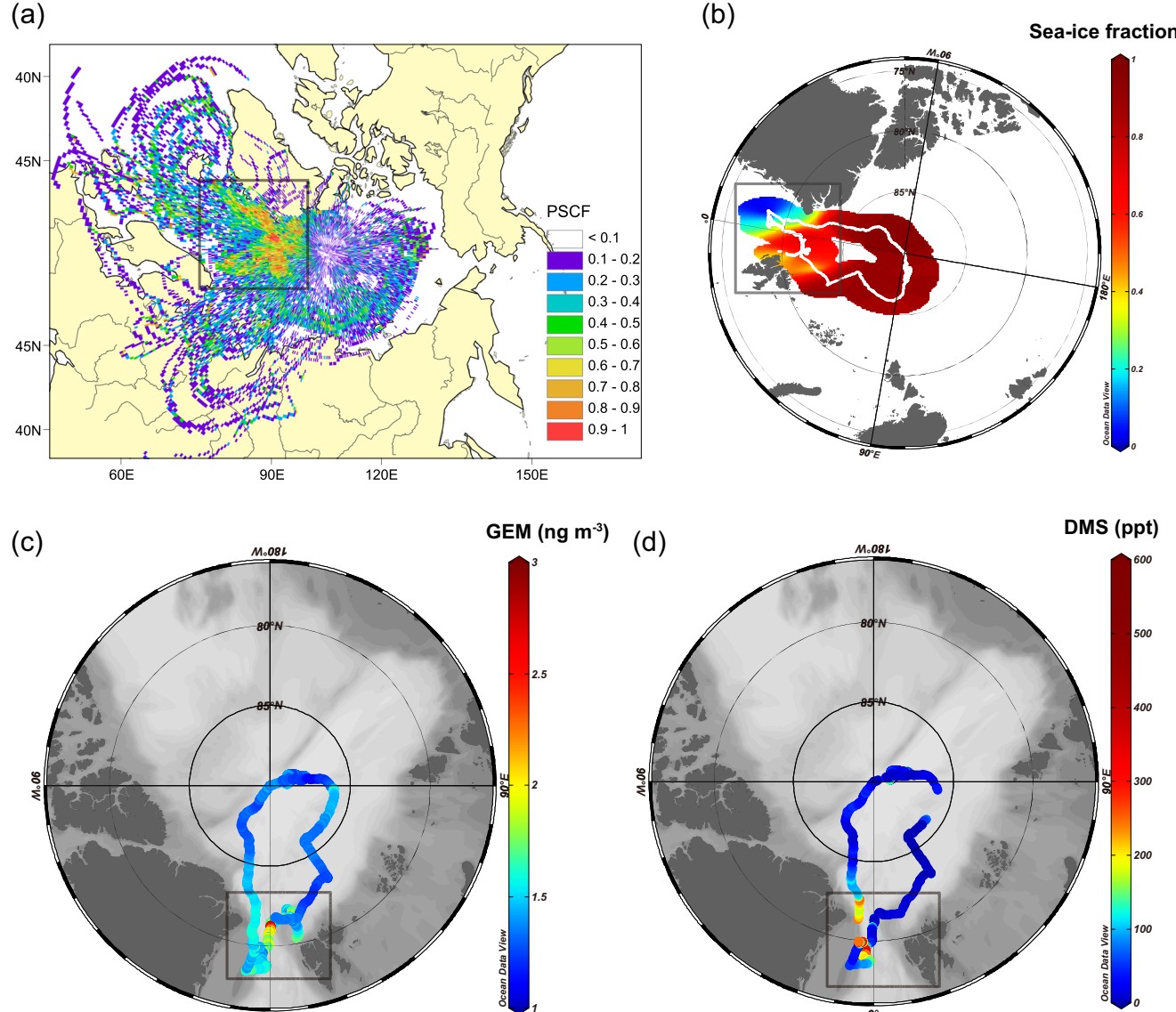

**Fig. 5 | Potential source contribution function (PSCF) analysis of gaseous elemental mercury (GEM) and spatial distribution of the sea-ice fraction, gaseous elemental mercury (GEM) and dimethylsulfide (DMS) during the observation period.** a Potential source contribution function (PSCF) analysis of gaseous elemental mercury (GEM) during the observation period (June 9–September 30, 2020) and (b) spatial distribution of the sea -ice fraction during the observation period. The black boxes in (a) and (b) indicate the region with high PSCF values (>0.6). The spatial distribution patterns of (c) GEM and (d) DMS. The black boxes in (c) and (d) mark high values of GEM and DMS that occurred in similar low-latitude areas during this cruise. a was generated using Meteoinfo software (http://www.meteothink.org/)[86]. b–d were generated using Ocean Data View (https://odv.awi.de/)[85].

## Implications in the context of rapid Arctic warming

Based on the unique and continuous set of GEM observations during the MOSAiC expedition, we show that oceanic evasion is an important source of atmospheric Hg in the central Arctic summertime. This study further verifies our current understanding of the central Arctic Hg cycle, specifically regarding the source of the summertime GEM maximum and the general role of the Arctic Ocean as a Hg source during the summertime[8,24]. Furthermore, this study offers new insights by showing that oceanic evasion is not significant throughout the central Arctic Ocean but mainly occurs in the MIZ. Our estimate of the Hg evasion flux suggests a higher magnitude than in the open ocean, causing a rapid increase of GEM concentrations throughout the Arctic.

The Arctic is warming faster than the rest of the planet and Arctic sea-ice is declining rapidly with potential consequences for the Hg cycle[60]. Firstly, projected future Arctic warming might favor conditions that stimulate AMDEs· driven Hg deposition through enhanced reactive halogen sources[8]. Long-term (1996–2017) observations of BrO· vertical column densities (VCDs) show an increasing trend of about 1.5% of the tropospheric BrO· VCDs per year during polar spring, which may be attributed to the increase in first-year ice coverage that has a higher salinity than multiyear ice and facilitates the production of BrO[61]. This could enhance the deposition of Hg and thus the load of Hg in surface seawater during the melt season. Secondly, the width of the MIZ in the warm season (July–September) in the central Arctic Ocean has increased by 13 km decade$^{-1}$ from 1979 to 2011, driven by the decrease of Arctic sea-ice extent and the replacement of thick, multi-year ice with thin, first-year ice[62]. In this context, it can be expected that the retreat of multi-year ice and the decline in sea-ice extent will continue, causing further increase in the width of the MIZ. This would translate into a larger source area of GEM in summertime. Thirdly, the decline in sea-ice extent and expansion of the MIZ can (1) transmit

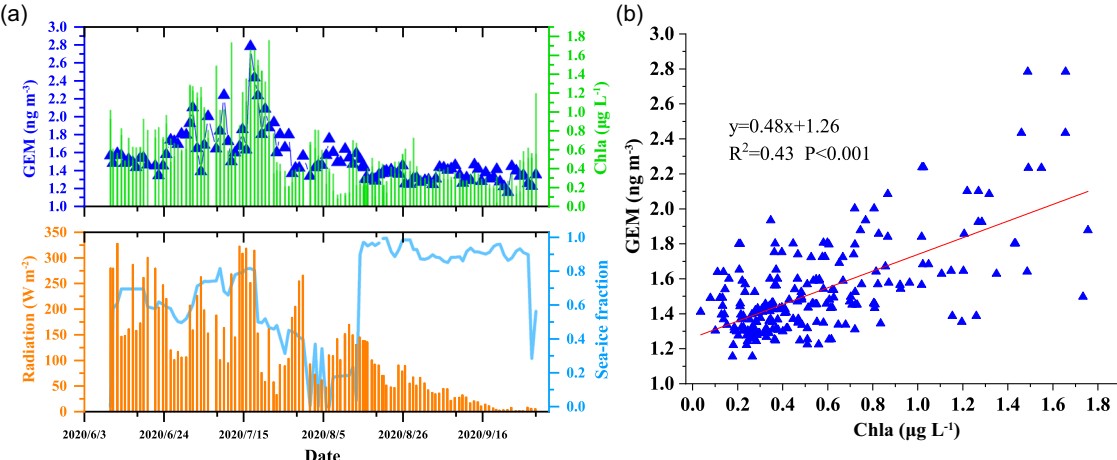

**Fig. 6 | Variation characteristics and their relationship of gaseous elemental mercury (GEM) and several environmental parameters. a** Time series of gaseous elemental mercury (GEM), chlorophyll-a (Chla), solar radiation (radiation) and sea-ice fraction. **b** The linear correlation between GEM and Chla.

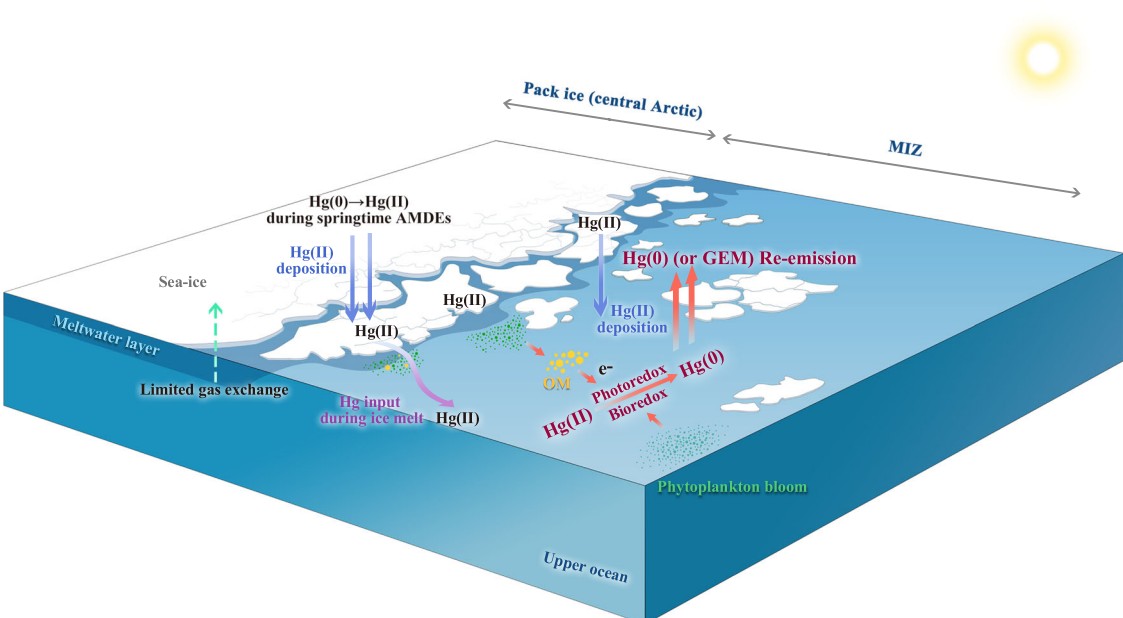

**Fig. 7 | Diagram of the main processes that drive the significant Hg(0) or gaseous elemental mercury(GEM) (re)-emission in the Marginal Ice Zone (MIZ).** The main processes including (1) previously deposited Hg(II) during springtime atmospheric Hg depletion events (AMDEs) is transferred to the surface ocean with melting ice/snow-water, resulting in a large Hg load in surface seawater in the MIZ; (2) high phytoplankton mass in the MIZ may lead to a high reduction capacity for Hg(II) in the MIZ, as the excretion of photoreactive organic compounds by phytoplankton can facilitate the photoreduction of Hg(II) to Hg(0) via electron transfer. In addition, phytoplankton can also directly contribute to the reduction of Hg(II) to Hg(0) through an enzymatic detoxification mechanism linked to their photosynthetic activity; and (3) the melting of sea-ice and the absence of upper ocean stratification in the MIZ (i.e., absence of a meltwater layer) facilitates air–sea gas exchange. The figure was generated using Adobe Photoshop 2021 software (version22.5).

substantially more light into the underlying water column and make a longer growing season, leading to an increase in phytoplankton production[63–65], and (2) promote $CO_2$ uptake by open seawater and the biologic carbon pump process in the MIZ[66,67], which would further increase the photo- and/or biological reduction capacity of Hg(II) there. As a result, we expect that in a context of Arctic warming, oceanic Hg evasion in the MIZ might become increasingly significant, and strengthen the role of the Arctic Ocean as a summertime Hg source. Given the relatively long atmospheric lifetime of GEM, this may have global consequences on the Hg cycle.

## Methods
### Study site
We refer here to measurements performed during the summertime legs of the year-long Multidisciplinary drifting Observatory for the Study of Arctic Climate (MOSAiC) expedition, from June 9 to September 30, 2020. The geographical range of this study is 78.34–90°N, 40.93°W–175.7°E, and the drift track is displayed in Fig. 1. Refer to Supplementary Fig. 9 for more information regarding the spatio-temporal position of *Polarstern*. More information on the atmospheric observations during MOSAiC can be found in ref. 68.

## GEM measurements

GEM was continuously measured using a cold vapor atomic fluorescence spectroscopic (CVAFS) Hg analyzer (Tekran™ 2537B). The sampling inlet of the Hg analyzer was mounted at the front of the Research Vessel *Polarstern* to minimize the influence of the ship exhaust. The flow rate of the sample air was set as 0.7 L·min⁻¹, and Hg in ambient air was alternatively trapped on two gold cartridges with a cycle time of 5 min. The trapped Hg on the cartridge was then thermally desorbed at 550 °C and detected by CVAFS. The inlet system was equipped with two soda lime tubes and two 0.45 μm Teflon filters to remove moisture and coarse particles. The observed data were carefully screened for local contamination as discussed in Angot et al.[69] and Beck et al.[70]. Calibration using a built-in internal Hg permeation source was conducted on a regular basis. In addition, external calibration through manual injections of known GEM contents using a Tekran 2505 unit was performed before and after the cruise, and the accuracy of both calibrations was better than 95%. The detection limit (DL) for GEM measurements using the 2537B unit is lower than $0.10 \text{ ng·m}^{-3}$. In addition, GEM measurements were also independently performed in the University of Colorado (CU) sea-laboratory container during the whole expedition[69,71]. Briefly, a Tekran 2537B analyzer was used to analyze 15-min integrated samples. Millipore 0.45 μm polyether sulfone cation-exchange membranes were used to remove Hg(II) species, and the instrument was automatically calibrated every 25 h. Data for this analysis were intercompared and adjusted for calibration offset using CU measurements as reference.

## Other ancillary data

We use DMS, CO, and $SO_2$ datasets collected during the MOSAiC expedition as ancillary data in this analysis. These datasets are described by ref. 69. DMS measurements were conducted using an Atmospheric Pressure Ionization Mass Spectrometer with an Isotopically Labeled Standard (APIMS-ILS)[72]. This instrument monitors the DMS mole fraction of a dried sample air stream at 10 Hz, averaged to 10 s, and the detection limit is typically <5 ppt[73]. CO measurements were performed in two independent containers on the D-deck. Here, we use hourly averaged merged datasets that combine the cross-evaluated independent measurements[74]. $SO_2$ measurements were performed in the Swiss container on the D-deck of Research Vessel Polarstern using a Thermo Fisher Scientific instrument (model 43i)[75].

Furthermore, surface ocean chlorophyll a (Chla) was measured daily from the ships underway system (11 m depth). Two liters of seawater were filtered onto GF/F filters (Whatman®) in duplicate or triplicate and frozen at −80 °C until further analyses at the Alfred Wegener Institute in Germany. Samples were extracted at 4 °C in 90% acetone overnight and subsequently analyzed on a fluorometer (TD-700; Turner Designs, USA), including an acidification step (1 M HCl) to determine phaeopigments.

Navigation parameters and a subset of meteorological data, including wind speed (WS), surface air pressure (P) and air temperature (Temp), were acquired from the ship's scientific data system. Shortwave radiation (RD), ocean fraction (ocean coverage fraction per grid cell) and sea-ice fraction data were extracted from Goddard Earth Observing System-Forward Processing (GEOS-FP) assimilated hourly meteorological data (with a horizontal resolution of 2° × 2.5°). The open-water fraction, which represents the fraction of oceanic region without sea-ice cover in the corresponding 2° × 2.5° grid along this cruise, was calculated as given below:

$$\text{Open-water fraction} = \text{Ocean fraction} - \text{Sea-ice fraction} \qquad (1)$$

Finally, the HYSPLIT transport and dispersion model from NOAA-ARL (Air Resources Laboratory)[76] was used to generate 168 h air mass back-trajectories during the whole observation period to conduct further statistical analyses of air masses in this study (see Sections 2.3)[74].

## Evaluating the relative importance of various influencing factors

To evaluate the relative importance of selected controlling factors to observed GEM variation during the summertime legs, we used a generalized additive model (GAM), a flexible semi-parametric statistical model that is data-driven and accounts for both linear and nonlinear parameters. GAM combines diverse dependent variable connection functions (e.g., binomial distribution, Poisson distribution, exponential distribution) and additive assumptions and fits the complex relationship between the dependent variable and independent variable by adding different functions. GAM has been applied in the prediction of atmospheric Hg concentrations in urban areas[36,77] and to quantify the anthropogenic, ecological, and biogeochemical drivers of Hg levels in Pacific Ocean tuna[78]. This prediction was conducted using the "mgcv package" in R[36]. The GAM model can be described by the following equation:

$$g(\mu) = f_1(x1) + f_2(x2) + \cdots + f_n(xn) + \varepsilon \qquad (2)$$

where xi (i = 1, 2, 3,..., n) are various parameters as predictors and *fi* is the smooth function of the predictors. Here, the smooth functions were set as penalized cubic regression splines to ensure a balance between overfitting and underfitting the observed data by choosing the effective number of degrees of freedom; ε is the residual; μ is the expected dependent variable; and g is the link function that specifies the relationship between the nonlinear formulation and the expected variable. Here, the "identity link" function with Gaussian was used since the distribution of GEM concentrations fit a Gaussian distribution.

For the selection of predictors for GAM, the Akaike Information Criterion (AIC), the F values and the $R^2$ values were considered to ensure the effectiveness of each input variable. With the addition of each variable, higher fitting $R^2$ values, which indicate better fitting results, higher F values indicate higher sensitivity and relative importance of the added variable, along with lower AIC values which indicate proper parameter selection. Based on this method and to include the main potential factors affecting GEM as much as possible, 3 categories of predictors, including 6 parameters among the 10 tested parameters (Supplementary Table 1), were considered in this study, as they met the criterion of the AIC evaluation (Supplementary Fig. 10), and the contributions of the 6 parameters in the models were all significant ($p < 0.001$):

(1) Long-range transport of anthropogenic emissions of Hg: indicated by the observed CO mixing ratios and the distance of the endpoint of each 48 h trajectory (hourly resolution) from the corresponding cruise observation location (Traj48h);
The main anthropogenic sources of Hg in the Arctic include coal-fired power plants, nonferrous metal smelters, and resource extraction activities (e.g., mining, oil, gas)[8]. These industries can also emit a large amount of CO. Therefore, CO can be used as a tracer of anthropogenic Hg emissions[77,79,80].

(2) Local oceanic emissions: indicated by the open-water fraction (Open-water) at the Polarstern position.

(3) Meteorological factors: wind speed (WS), surface air pressure (P) and air temperature (Temp);

Previous studies suggested that GAM would be relatively robust when the adjusted $R^2$ value is greater than 0.5[36,77]. The GAM simulation result in this study shows that these variables could explain 63.3% of the variance in GEM concentration, with a fitting $R^2$ of 0.63. The GAM can well capture the variability of GEM during the whole observation

period, which indicates good performance in this study (Supplementary Fig. 11).

## Model validation

We further systematically evaluated the quality of the GAM using different methods. We used a 5-fold cross-validation test to assess the accuracy of GAM simulations. The principle of this method is to randomly divide the whole dataset into 5 subsets, and in each cross-validation round, four subsets are used to fit the model, and the remaining subset is used to make a prediction. This process was repeated 5 times to ensure that every subset was tested[77,81]. The 5-fold cross-validation results displayed a good coincidence between the GAM and cross-validated results (slope = 1.00, $R^2$ = 0.99), demonstrating good reliability of the model (Supplementary Fig. 11)[82]. In order to test the underlying assumptions of homogeneity, normality, and independence and ensure the validity and accuracy of the model, we used the following methods: (1) quantile-quantile (Q-Q) plot (sample quantiles against theoretical quantiles), (2) scatterplots of residuals against linear predictors, and (3) histograms of the residuals[36,77]. The Q-Q plot result showed that GAM produced good results around the average concentration; the scatterplot of residuals vs. linear predictor showed that residuals were generally concentrated around a value of 0, and presented a random distribution with no obvious trend, indicating unbiased simulations of GEM. The histogram of residuals was close to a normal distribution, which suggests that the error of model fitting is random and that the selection of predictor variables is reasonable (Supplementary Fig. 10). These results suggest that the selected predictors can be used to reliably identify sources of Hg in the central Arctic atmosphere.

## PSCF analysis

A potential source contribution function analysis (PSCF) was utilized in this study to identify the potential source regions of GEM in the Arctic Ocean during the observation period. This method is driven by Global Data Assimilation System (GDAS) meteorological data (with 1° × 1° latitude and longitude horizontal spatial resolutions and 23 vertical levels) and combines the observed hourly GEM concentrations with HYSPLIT backward trajectories. To cover the central Arctic Ocean (potential source region) to a maximum extent, 168-h HYSPLIT backward trajectories with GDAS data for every hour were used here. The arrival altitude of the trajectory was set as 50 m to match the altitude of the sampling site. The domain of 168-hour backward trajectories (57–90°N and 180°W–180°E) was divided into grid cells with a 0.5° × 0.5° resolution. A higher value of $PSCF_{ij}$ indicates a higher probability that a given region contributed to elevated GEM levels at the receptor site. The calculation of the $PSCF$ value for the $ij_{th}$ cell is shown below:

$$PSCF_{ij} = \frac{M_{ij}}{N_{ij}} \times W_{ij} \qquad (3)$$

where $M_{ij}$ is the number of trajectory segment endpoints in a cell associated with GEM concentrations higher than the overall mean and $N_{ij}$ is the total number of trajectory segment endpoints in a grid cell. The empirical weight function $W_{ij}$ was taken into consideration in the calculation to reduce the uncertainties of grid cells with small $N_{ij}$ values, and its values can be found in ref. 36. It should be noted that 57.3% of the trajectories' heights were <200 m, 81.1% <600 m, and 91.2% <1000 m (Supplementary Fig. 12), indicating that the PSCF results in this study mainly characterize atmospheric transport in the low troposphere.

## Data availability

The GEM and the corresponding meteorological (including air temperature, atmospheric pressure, solar radiation and wind speed) and hydrologic data (including Open-water fraction, sea-ice fraction) of this study have been merged and are available at figshare (https://doi.org/10.6084/m9.figshare.23614494)[83]. $SO_2$ data is available at https://doi.org/10.1594/PANGAEA.944270. CO data is available at https://doi.org/10.1594/PANGAEA.944389.

## Code availability

The R code of generalized additive model (GAM) used in this study is available at https://github.com/yfg66/GAM_Hg. The PSCF method and the corresponding software used in this study can be accessed here: http://www.meteothink.org/.

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

## Acknowledgements

Data reported in this manuscript were produced as part of the international Multidisciplinary drifting Observatory for the Study of Arctic Climate (MOSAiC) expedition with the tag MOSAiC20192020, with activities supported by Polarstern expedition AWI_PS122_00 and with the help of many people[84]. We thank the *China Arctic and Antarctic Administration* for fieldwork support. Some observations used here were provided by the Atmospheric Radiation Measurement (ARM) Climate Research Facility, a US Department of Energy (DOE) Office of Science User Facility sponsored by the Office of Biological and Environmental Research. We thank the many ARM operators who supported the field observations. We thank Hui Kang, Jianfang Chen and Songtao Ai for help in field work. We thank Weihua Gu, Hongwei Liu, and Lei Zhang for helpful discussion. This research was funded by the National Natural Science Foundation of China (grant no. 41941014), the US National Science Foundation (awards OPP 1807496, 1914781, and 1807163), the Swiss National Science Foundation (grant 200021_188478), the Swiss Polar Institute (grant DIRCR-2018-004), the DOE Atmospheric System Research Program (DE-SC0019251), and NOAA Physical Sciences Laboratory (NA22OAR4320151). JS holds the Ingvar Kamprad chair for extreme environment research, sponsored by Ferring Pharmaceuticals.

## Author contributions

Z.Q.X., F.G.Y., and H.A. conceived the study. Z.Q.X., F.G. Y., H.A., B.B., L.Y.Z., J.R., H.L.L., R.B.L., and M.D.S. performed the field investigation. C.J.M.H. participated in the campaign and provided the Chla data. J.S., I.B., D.Ho., T.J., T.L., L.Q., M.B., T.P., S.A., L.B., D.He., J.H., H.W.J., and K.P. participated in the campaign and provided ancillary data (including DMS, CO, and $SO_2$). F.G.Y. led the original paper writing and methodology with input from Z.Q.X., H.A., B.B., and J.S. All authors contributed to data interpretation and writing.

## Competing interests

The authors declare no competing interests.

## Additional information

[1]Institute of Polar Environment & Anhui Key Laboratory of Polar Environment and Global Change, Department of Environmental Science and Engineering, University of Science and Technology of China, Hefei, Anhui 230026, China. [2]Extreme Environments Research Laboratory, École Polytechnique Fédérale de Lausanne (EPFL) Valais Wallis, Sion, Switzerland. [3]Institute for Arctic and Alpine Research (INSTAAR), University of Colorado Boulder, Boulder, CO, USA. [4]Univ. Grenoble Alpes, CNRS, INRAE, IRD, Grenoble INP, IGE, 38000 Grenoble, France. [5]Cooperative Institute for Research in Environmental Sciences, University of Colorado, Boulder, CO, USA. [6]NOAA, Physical Sciences Laboratory, Boulder, CO, USA. [7]Alfred Wegener Institut—Helmholtzzentrum für Polar- und Meeresforschung, Am Handelshafen 12, 27570 Bremerhaven, Germany. [8]Key Laboratory for Polar Science of the MNR, Polar Research Institute of China, Shanghai, China. [9]Third Institute of Oceanography, Ministry of natural resources, Xiamen, China. [10]Key Laboratory of Marine Ecosystem Dynamics, Second Institute of Oceanography, Ministry of Natural Resources, Hangzhou 310012, China. [11]School of Oceanography, Shanghai Jiao Tong University, Shanghai 200030, China. [12]Institute for Atmospheric and Earth System Research (INAR)/Physics, Faculty of Science, University of Helsinki, Helsinki, Finland. [13]Climate & Atmosphere Research Centre (CARE-C), The Cyprus Institute, Nicosia, Cyprus. [14]Bigelow Laboratory for Ocean Sciences, Boothbay, ME, USA. [15]Boulder Atmosphere Innovation Research, Boulder, CO, USA. [16]JH Atmospheric Instrumentation Design, Boulder, CO, USA. ✉e-mail: helene.angot@univ-grenoble-alpes.fr; zqxie@ustc.edu.cn

