## [Peer Review File · Nature Communications]

The Marginal Ice Zone as a dominant source region of atmospheric mercury during central Arctic summertimeReviewer #1 (Remarks to the Author):

SUMMARY

This manuscript presents continuous measurements of summertime gaseous elemental mercury (GEM) in the central arctic, made during the MOSAiC expedition. The authors modeled the importance of oceanic evasion as the dominant source of GEM to the Arctic atmosphere. The data and modeling presented have important implications in the distribution and deposition of Hg in Arctic environments, and provide further evidence in support of trends in dissolved gaseous mercury (DGM) and GEM from earlier cruises in the Arctic.

The authors provide significant data analysis and interpretation, originating from sound analytical methodology. Overall, I found the manuscript very well written, making an important contribution to the field stemming from a unique opportunity (MOSAIC). I only have a few comments, with my main concern dealing with the explanation of high GEM levels in the marginal ice zone (MIZ).

TECHNICAL COMMENTS

Line 189 – You state that transport of anthropogenic emissions to high latitudes is less likely in the summertime, but also state that the Arctic dome, which acts as a transport barrier, recedes in the summer. Wouldn't this make transport northward more feasible, or am I misunderstanding your reasoning?

Section 2.3.2 second point – Line 235-253 – I don't disagree with/dispute your observed correlation between Chla and GEM, and you present multiple hypotheses to explain the trend, I just might be a little more careful about overstating a necessity for correlation to imply causation. Is it necessary for the increased GEM measured in the MIZ to be the result of DGM produced in the MIZ? Or might it have just been released once the sea ice was no longer present, which also happens to be where phytoplankton bloom. Previous studies measuring DGM in the Arctic show high concentrations under contiguous ice, not just in the MIZ, and I'm not sure you addressed/discussed this sufficiently in the manuscript.

EDITORIAL COMMENTS

I think the abstract highlights the role of observations made in the conclusions reached, but could do a better job highlighting the major role that the modeling tools (e.g., generalized additive model (GAM), potential source contribution function analysis (PSCF), air-mass back trajectories) played in this manuscript.

Line 91-96 – the sentence could be broken up into two separate sentences

Some abbreviations need to be defined earlier – eg, GAM (line 158), PSCF (line 209). I also believe the abbreviation RD is only defined in the supporting info.

Reviewer #2 (Remarks to the Author):

Levels of atmospheric gaseous elemental mercury (GEM) exhibit a summertime maximum in the Arctic, however, the source of this peak is not well understood. The source of the summertime GEM maximum is debated. This paper presents an analysis of GEM concentrations and their variability in the central Arctic Ocean based on data collected during the MOSAiC international expedition, with a focus on identifying sources contributing to the summertime GEM levels in the region. To identify the source(s), the study used a GAM simulation.

The study found that long-range transport of anthropogenic and terrestrial Hg from lower latitudes is a minor contribution (~2%), while the dominant source (>50%) is oceanic evasion and regional transport. The study also suggests that oceanic evasion

mainly occurs in the Marginal Ice Zone, which could be a significant cause of the summertime GEM maximum in the Arctic.

The manuscript is easy to read and does a good job of discussing the possible origin of the summertime GEM maximum by addressing the relative contributions from (1) the long-range transport of anthropogenic emissions, (2) local oceanic emissions and (3) meteorological conditions. The measurements collected in this study are significant, largely contributing the current lacking dataset of GEM in central Arctic Ocean. The auxiliary data is also considerable and helps support the authors interpretations.

The results of the study are mostly clear, however, the overarching conclusions could be better highlighted. The study would benefit of emphasizing more the implications of their results. For example, if ocean evasion and regional transport are the main source of summertime GEM maximum, and if ocean evasion mainly occurs in the Marginal Ice Zone, what are the implications of ocean evasion being mainly in the MIZ relative to the open ocean? Other studies could be referenced to support the findings reported here. I would suggest describing how the results of this study compare (or don't) with the reported observations from coastal stations. The authors reference a few studies, but there have been more. I recommend the authors draw on the work done by others to help support their conclusions.

The authors mention that long-term observations from coastal stations may provide an incomplete understanding of the atmospheric Hg cycle in the central Arctic and lead to uncertainties in modeling atmospheric Hg in that region. This is important, how do the results from this study fill that gap? Finally, the big picture implications of the results are somewhat lacking. The manuscript would benefit from providing some discussion of the results in context of a changing climate, particularly with respect to a warming Arctic.

Overall, I think this manuscript will contribute to the Hg research community and hopefully at a larger scale.

Line by line comments:

- Line 67 – two references here but only refers to the AMAP report
- Line 73 – May want to provide some context, definition to AMDEs, since the first time introduced
- Line 78 – Word choice "unreveled"...revealed? Suggests this has been solved, yet the summertime maximum is still debated
- Line 85 – There are other important isotope studies to consider here, Douglas and Blum (2019); Zheng et al. (2021)
- Line 89 – May be good to provide reference here to the coastal studies
- Line 135 – define RD, might be worth indicating that these are used as a proxy for anthropogenic emissions
- Line 139 – other studies could be referenced here as well
- Line 149 – check section reference (numbering is wrong?)
- Line 151 – this is interesting. Could authors provide reasoning as to why? Could it have occurred earlier in the year?
- Line 156 - May be good place to note the significance of the summertime GEM maximum and knowing its origin
- Line 216 – additional references needed here
- Line 231 - This is somewhat confusing because above the authors suggest AMDEs were not observed.
- Line 282 – somewhat vague statement. Also what about temperature on atmospheric chemistry (e.g., increasing temperature on emission of halogens/ oxidants, AMDEs, deposition of oxidized Hg to snow, regionally, MIZ as major sinks, etc.). Authors could draw on other studies here.
- Line 285 to 289 – reference to help support explanation.
- Line 294 – it is not clear why WS outside of the range (at lower and higher ends of the

range) would promote sea-air exchange and regional transport and why this specific range implies dilution. May need more clarity.

Methods: What a sensitivity analysis completed on the model used? This can be added to supplemental information.

Figures:

Some figures are fuzzy and difficult to read. Suggest higher resolution images (e.g., 4a, 4b, 5b)

Reviewer #3 (Remarks to the Author):

General comments:

The study presents unique atmospheric GEM concentration data from the Arctic Ocean. The data were obtained using a Tekran 2537B system running on board of the research vessel Polarstern from June to September 2020. The study confirms an offshore summertime GEM peak between Greenland and Svalbard, a GEM maximum that has been observed at several coastal Arctic monitoring stations. The peak is not associated with atmospheric long-range transport of Hg. Instead a GAM simulation indicates that open ocean evasion of GEM, in particular from the low-latitude marginal ice zone between Greenland and Svalbard, is a significant source of GEM to the Arctic atmosphere in summer. While I want to point out the originality of the GEM dataset, I have some concerns regarding the identification of sources contributing to the GEM summertime peak.

The authors report that 63.3% of the summertime GEM variation can be largely explained by open water re-emission and meteorological factors. Araujo et al., 2022, Nat. Commun. evidenced that the GEM peak originates from re-emission of Hg deposited to snow and sea ice during AMDEs. They concluded that GEM re-emission takes place directly from snow and sea ice but also from regional open waters that receive meltwater Hg inputs. How much of the GEM variation in the present study can be explained by direct GEM re-emission from snow and sea ice? Is re-emission from snow and sea ice not contributing to the GEM summertime peak in the central Arctic Ocean at all? Please include a paragraph in the ms about how direct snow and sea ice re-emission impacts GEM variation in summer.

Long-range transport of Hg contributes only marginally to the GEM summertime peak. Meteorological factors, however, explain 37% of the identified GEM variation in the central Arctic Ocean. Advection of Hg from the mid-latitudes (i.e. long-range transport of anthropogenic emissions and terrestrial Hg) and Hg evading from the ocean can indeed be considered sources contributing to GEM levels in the Arctic atmosphere. However, please explain how meteorological factors can be a source of GEM. Meteorological factors such as wind speed or solar irradiation have been demonstrated to influence the rate of GEM evasion from the ocean but cannot be considered a source in my understanding.

In July 2020, when the GEM peak was detected, the Polarstern was cruising between Greenland and Svalbard, correct? I wonder whether the July GEM peak could have been detected close to the North Pole as well? Did you find any indication for that? Please add dates and the respective location of the Polarstern in Fig. 1a.

Overall, ocean evasion explains less than 50% of the GEM variation in summer (L404: 36.7 % of the GEM variation cannot be explained). Still, there is strong indication for large GEM evasion from the MIZ between Greenland and Svalbard. It would strengthen the manuscript, if the authors could include a statement about the GEM flux rate ($\text{ng m}^{-2} \text{ day}^{-1}$) that could be expected from the MIZ between Greenland and Svalbard. What flux rate would be required to increase the GEM concentration within the Arctic Ocean boundary layer from 1.35 ng m^{-3} (September, background) to 1.8 ng m^{-3} (July,

summertime peak). Under the assumption that GEM re-emission from the other parts of the central Arctic Ocean remain small.

Such a statement could be integrated into a yet missing "conclusion" or "implication" paragraph. I strongly suggest to add conclusions and implications at the end of the main text. Please discuss the significance of your results and how your findings fit with our current understanding of the Arctic mercury cycle (cp. with Dastoor et al., 2022, Nature reviews, earth and environment and Araujo et al., 2022, Nat. Commun.).

Specific comments:

Line 62: There are more recent studies reporting atmospheric lifetimes of GEM. Consider referring to Saiz-Lopez et al., 2018, Nat. Commun. or Horowitz et al., 2017, ACP.

Line 67: The AMAP report is referenced with number 44 instead of 8. Is there a reason for that? Adjust reference numbering throughout the ms.

Line 90: Delete "unfortunately".

Line 135: Write out "RD". You define all abbreviations in the Method section. Please do that in main text instead.

Line 194: "Central Arctic Ocean" or "central Arctic Ocean"? Be consistent.

Line 218: What would be an important factor? This sentence is hard to understand. Please rephrase.

Line 236: Write out "Chla".

Line 270: Write out as follows: Wind speed, atmospheric pressure and air temperature. Stick to "re-emission".

L313: The flow rate of the Tekran 2537B was 0.7 L min⁻¹ which is low compared to most Tekran deployments. Is there a reason why the authors used such a low flow rate? Analysis of GEM in 3.5 L of air is at the lower limit. Please explain briefly why an additional Tekran analyzer was setup even though GEM was determined in the CU container on a 15 min resolution? Please also add a comment about how you avoided to measure GEM that potentially originated from ship exhaust gas emissions.

L 685, 697, 709: Add units to the axis text in Figs. 3a,b, 4a,b, 5b.

Supporting information:

L 38: Is that a table of content? Please adjust.

L 45: Why is the section title number 3.2? Please adjust.

L 70: How was this figure created? Using HYSPLIT? Information is missing in the caption.

L 95: Add units to the axis text in Fig S7.

Reviewer #4 (Remarks to the Author):

This study investigated key mechanisms for occurrence of the summer concentration peak of gaseous elemental mercury (GEM) in the Arctic based on observations during the Multidisciplinary drifting Observatory for the Study of Arctic Climate (MOSAIC)

expedition and a statistical modeling approach. Oceanic evasion in the Marginal Ice Zone was found to be the dominance source for the summertime GEM maximum, and the photoreduction of Hg(II) to GEM by phytoplankton was proposed to be the main mechanism. The observational data obtained in the study are valuable, and the finding reported in the study is interesting which could draw widespread attention. However, the generalized additive model (GAM) used in this study needs more information on model validation, and more solid evidence needs to be provided for the mechanisms the authors proposed. A comprehensive schematic diagram for what's going on during the summer GEM maximum is encouraged to be added to the manuscript for readers to get a whole picture of the crucial mechanisms. Overall in my opinion, substantial improvement is required before the manuscript is acceptable for publication on Nature Communications. Here are some specific comments:

1. **Abstract:** The most probable cause of the summer GEM peak, the photoreduction of Hg(II) to GEM by phytoplankton (if I understand the Results and Discussion part correctly), is not reflected in the abstract. The authors only mentioned that oceanic evasion mainly occurs in the Marginal Ice Zone, which could mislead the readers to think that the air-ice Hg exchange might be the dominant mechanism of the summer GEM maximum.
2. **Line 82:** The citation should be Durnford et al. according to the reference list.
3. **Lines 82–87:** The summary of the GEM source apportionment results could cause misinterpretation. Long-range transport of Asian air is a primary source. Legacy Hg emissions or re-emissions from terrestrial regions or ice zones are considered as secondary sources. Due to the different methodologies adopted, a certain part of the discrepancy between these studies could be caused by the categorization of GEM sources. Some legacy Hg from historical Asian air could be accumulated in plants, snowpack and ice in the Arctic area and reemitted as a secondary source of GEM. It would be helpful for the readers to understand the Arctic Hg cycle if the authors could make a more comprehensive introduction here, incorporating these literatures.
4. **Line 116:** Misspelling of "coastal".
5. **Section 2.1:** The discussion in this section is limited. Significance of difference from previous studies based on t-test should be given. The standard deviation of the central Arctic GEM observations during August 25–September 8, 2012 (0.61) is much higher than that in this study (0.07). What caused the large fluctuation in the study of Yu et al. (2014)?
6. **Lines 149–151:** What could be the reason for the absence of this phenomenon in the MOSAiC expedition (especially in June)? Why did the authors present this comparison here?
7. **Figure 4c:** What is the significance of the relationship between land fraction and GEM? The negative correlation seems not quite robust to me.
8. **Lines 182–194:** The reasoning is a bit farfetched. More solid evidence should be provided to support the speculation.
9. **Figure 5b:** What do these three peaks stand for in this influencing pattern? This is an important question to verify the reliability of the GAM model.
10. **Line 230:** How much Hg(II) is stored yearly approximately? More quantitative comparisons are required.
11. **Line 239:** It should be Figure 7.
12. **Section 2.3.3:** Solar radiation should be included as a meteorological predictor in the GAM model. In the previous section, the photoreduction of Hg(II) to GEM by phytoplankton is proposed to be a dominant mechanism. If so, solar radiation is supposed to be a key factor for the summertime GEM maximum.

Responses to Reviewers' Comments

Dear Reviewers:

Thank you for your comments on our manuscript entitled “The Marginal Ice Zone as a dominant source region of atmospheric mercury during Arctic summertime” (manuscript ID: NCOMMS-23-05574) and for offering us an opportunity for revision. We believe the constructive comments have helped us improve the overall quality of the manuscript. Below are our detailed responses to reviewers' questions. Revised texts of the manuscript are marked in red in the responses to reviewers' comments.

Reviewers' Comments and response

Reviewer #1:

This manuscript presents continuous measurements of summertime gaseous elemental mercury (GEM) in the central arctic, made during the MOSAiC expedition. The authors modeled the importance of oceanic evasion as the dominant source of GEM to the Arctic atmosphere. The data and modeling presented have important implications in the distribution and deposition of Hg in Arctic environments, and provide further evidence in support of trends in dissolved gaseous mercury (DGM) and GEM from earlier cruises in the Arctic.

The authors provide significant data analysis and interpretation, originating from sound analytical methodology. Overall, I found the manuscript very well written, making an important contribution to the field stemming from a unique opportunity (MOSAiC). I only have a few comments, with my main concern dealing with the explanation of high GEM levels in the marginal ice zone (MIZ).

Q: Line 189 – You state that transport of anthropogenic emissions to high latitudes is less likely in the summertime, but also state that the Arctic dome, which acts as a transport barrier, recedes in the summer. Wouldn't this make transport northward more feasible, or am I misunderstanding your reasoning?

Re: Atmospheric conditions in winter and early spring are conducive to long-range transport from regions bordered by the Arctic front (Willis et al., 2018). Minimum values of potential temperature that occur in the Arctic lower troposphere, which isolate the near-surface Arctic from the rest of the atmosphere, can form irregular closed domes over Arctic regions with surfaces of constant potential temperature (isentropic surfaces) (Klonecki et al., 2003). In the absence of diabatic heating or cooling, air masses will travel adiabatically along isentropic surfaces and be forced to ascend as they move northward (Klonecki et al., 2003; Law and Stohl, 2007). Therefore, pollution emitted at low potential temperatures is much more likely to reach the lower Arctic troposphere than pollution emitted at higher potential temperatures on time scales up to several days (Iversen, 1984). The Arctic front can extend as far south as 40 °N in January (wintertime) (Willis et al., 2018). As a result, pollution emitted in cold regions north of the Arctic front (north of 40 °N, e.g., northern Eurasia) can be transported directly to Arctic regions in the lower troposphere during winter. In contrast, warm and moist air masses originating from North America and Asia are forced to ascend and can access the Arctic middle and upper troposphere rather than low troposphere (Law and Stohl, 2007; Willis et al., 2018). However, in summertime, the Arctic dome recedes poleward from approximately 40 °N in the winter to roughly north of 70 °N, such that direct transport from sources south of the Arctic front (south of about 70 °N, such as northern Eurasia with higher anthropogenic emissions) becomes much less important. Mean residence times within the Arctic increase during summer, particularly near the surface, illustrating the decreased tendency for long-range transport during summer compared to winter and spring. The seasonality of poleward pollution transport from mid-latitudes is well described in the literature, see for instance the recent publications by Bozem et

al. (2019) or Boyer et al. (2023). We have also added some corresponding explanation in the manuscript, which is also displayed below:

“**Line 217** (section 2.3.1): One potential explanation for the minor contribution of land-based anthropogenic/terrestrial Hg emissions to summertime GEM levels in the **central** Arctic is the position of the Arctic dome. The Arctic dome acts as a transport barrier for air masses outside the Arctic dome and isolates the Arctic lower troposphere from lower latitudes, driven by thermal stratification of the lower atmosphere within the polar dome (Klonecki et al., 2003). **The Arctic dome can extend to approximately 40°N in wintertime, therefore favoring the transport of mid-latitude Hg emissions, and recede poleward to roughly north of 70° N in summertime, isolating the central Arctic from mid-latitudes (Shaw, 1995; Willis et al., 2018). More information on the seasonality of poleward pollution transport from mid-latitudes can be found in Bozem et al. (2019) or Boyer et al. (2023).** This is also reflected in the 7-day backward trajectories, which show that air masses were mainly concentrated over the Arctic Ocean during the observation period (Figure S5). Moreover, the most recent AMAP/UNEP global Hg emission inventory shows few anthropogenic Hg sources in land areas around the Arctic Ocean (Dastoor et al., 2022b), which supports the low contribution of anthropogenic land-based Hg emissions to the **central** Arctic Ocean in this analysis.”

Q2: Section 2.3.2 second point - Line 235-253 - I don't disagree with/dispute your observed correlation between Chla and GEM, and you present multiple hypotheses to explain the trend, I just might be a little more careful about overstating a necessity for correlation to imply causation. Is it necessary for the increased GEM measured in the MIZ to be the result of DGM produced in the MIZ? Or might it have just been released once the sea ice was no longer present, which also happens to be where phytoplankton bloom. Previous studies measuring DGM in the Arctic show high concentrations under contiguous ice, not just in the MIZ, and I'm not sure you addressed/discussed this sufficiently in the manuscript.

Re: Thank you for raising this point. Indeed, DGM can accumulate under sea-ice and then be released when sea-ice disappears, and the presence of contiguous ice can be a barrier to the sea-air exchange of Hg and interfere with the evasion of the DGM, despite high DGM accumulated under contiguous ice (DiMento et al., 2019). Therefore, we think that the increased GEM observed in the MIZ is not only due to the DGM produced there, but also due to the environmental conditions, including sea-ice melt and the low upper ocean stratification in the MIZ, which, taken together, contribute to the evasion of DGM to air. We have also discussed this in the third point of Section 2.3.2. For the observed significant correlation between Chla and GEM, we agree that this does not mean there is a causation. Previous lab studies have shown that phytoplankton contributes to Hg(II) reduction. Our in situ observations support this hypothesis, which is worth reporting here. We have rephrased the statement here, which is also shown below. This does not change the main finding of the manuscript: ocean Hg evasion mainly occurs in the MIZ, compared with open sea and contiguous ice areas.

“Line 292: Several previous laboratory studies have suggested that the presence of marine phytoplankton can facilitate Hg(II) reduction to dissolved gaseous Hg (Ariya et al., 2015; Deng et al., 2009; Lanzillotta et al., 2004; Ben-Bassat et al., 1972). The observed significant positive correlation between GEM and Chla could support the hypothesis that biological activity, which is particularly high in the MIZ in summer, facilitates the production of GEM (or Hg(0)). Different potential mechanisms for such an interaction have been suggested in the past: (1) the excretion of photoreactive organic compounds by phytoplankton can facilitate the photoreduction of Hg(II) to Hg(0) via electron transfer (Ariya et al., 2015; Deng et al., 2009; Lanzillotta et al., 2004). This hypothesis may be supported by the occurrence of concurrent high solar radiation and high levels of particulate organic carbon (POC) observed in the surface seawater during the GEM and Chla peaks (Figure 7 and Figure S6). (2) Phytoplankton can directly contribute to the reduction of Hg(II) to GEM through an enzymatic detoxification mechanism linked to their photosynthetic activity (Ariya et al., 2015; Ben-Bassat et al., 1972). These mechanisms could explain why the MIZ has a high Hg(II) reduction capacity.”

Q3: I think the abstract highlights the role of observations made in the conclusions reached, but could do a better job highlighting the major role that the modeling tools (e.g., generalized additive model (GAM), potential source contribution function analysis (PSCF), air-mass back trajectories) played in this manuscript.

Re: Thanks for your suggestion. We have revised the abstract as follows:

“**Abstract:** Atmospheric gaseous elemental mercury (GEM) concentrations in the Arctic exhibit a clear summertime maximum, while the origin of this peak is still a matter of debate in the community. Based on summertime observations during the Multidisciplinary drifting Observatory for the Study of Arctic Climate (MOSAIC) expedition and a modeling approach, we further investigate the sources of Arctic atmospheric Hg. **Simulations with a generalized additive model (GAM)** show that long-range transport of anthropogenic and terrestrial Hg from lower latitudes is a minor contribution (~2%). The dominant source (> 50%) is oceanic evasion. **A potential source contribution function (PSCF) analysis** further shows that oceanic evasion is not significant throughout the Arctic Ocean but mainly occurs in the Marginal Ice Zone **due to the specific environmental conditions in that region. Our results suggest that this regional process could be the leading contributor to the observed** summertime GEM maximum. In the context of rapid Arctic warming and the observed increase in width of the MIZ, oceanic Hg evasion may become more significant and strengthen the role of the Arctic Ocean as a summertime source of atmospheric Hg.”

Q4: Line 91-96 – the sentence could be broken up into two separate sentences
Some abbreviations need to be defined earlier – eg, GAM (line 158), PSCF (line 209).
I also believe the abbreviation RD is only defined in the supporting info.

Re: Thanks for your comments. We have broken up the corresponding sentence into two separate sentences, which are also shown below:

“**Line 109:** For example, regional differences are expected for sea ice physical properties (e.g., dynamic deformation, thermal processes (melting and freezing)), and chemical composition. **These factors** are expected to have non-negligible impacts on

key processes of the Hg cycle, such as redox reactions and snow re-emission of Hg, ultimately affecting air-sea exchange and atmospheric Hg concentrations in the central Arctic (Durnford and Dastoor, 2011; Moore et al., 2014; Yu et al., 2014)”

We have also added the corresponding detailed definitions of the abbreviations in main text based on your comments.

Reviewer #2:

Levels of atmospheric gaseous elemental mercury (GEM) exhibit a summertime maximum in the Arctic, however, the source of this peak is not well understood. The source of the summertime GEM maximum is debated. This paper presents an analysis of GEM concentrations and their variability in the central Arctic Ocean based on data collected during the MOSAiC international expedition, with a focus on identifying sources contributing to the summertime GEM levels in the region. To identify the source(s), the study used a GAM simulation.

The study found that long-range transport of anthropogenic and terrestrial Hg from lower latitudes is a minor contribution (~2%), while the dominant source (>50%) is oceanic evasion and regional transport. The study also suggests that oceanic evasion mainly occurs in the Marginal Ice Zone, which could be a significant cause of the summertime GEM maximum in the Arctic.

The manuscript is easy to read and does a good job of discussing the possible origin of the summertime GEM maximum by addressing the relative contributions from (1) the long-range transport of anthropogenic emissions, (2) local oceanic emissions and (3) meteorological conditions. The measurements collected in this study are significant, largely contributing the current lacking dataset of GEM in central Arctic Ocean. The auxiliary data is also considerable and helps support the author’s interpretations.

Q1: The results of the study are mostly clear, however, the overarching conclusions could be better highlighted. The study would benefit of emphasizing more the implications of their results. For example, if ocean evasion and regional transport are the main source of summertime GEM maximum, and if ocean evasion mainly occurs in the Marginal Ice Zone, what are the implications of ocean evasion being mainly in the MIZ relative to the open ocean? Other studies could be referenced to support the findings reported here. I would suggest describing how the results of this study compare (or don't) with the reported observations from coastal stations. The authors reference a few studies, but there have been more. I recommend the authors draw on the work done by others to help support their conclusions.

Re : Thank you for your suggestions. We have added a sub-section titled "Implications in the context of rapid Arctic warming" (see also Q3 of your comments) to discuss the implications. Briefly, and as stated in the revised abstract, in the context of rapid Arctic warming and the observed increase in width of the MIZ, oceanic Hg evasion may become more significant and strengthen the role of the Arctic Ocean as a summertime source of atmospheric Hg.

We have also referred to more Arctic coastal studies:

"Line 232: Skov et al. (2020) reported a low contribution of anthropogenic emissions (14%-17%) to the GEM concentrations at Villum station in northern Greenland."

"Line 243: A previous study based on GEM data collected at Zeppelin station combined with the Lagrangian particle dispersion model FLEXPART also found that the highest GEM concentrations in July and August were associated with air masses from the marine boundary layer, suggesting the likely contribution of oceanic Hg evasion (Hirdman et al., 2009)."

"Line 261: In addition, multi-year observations of GEM at several Arctic coastal stations showed that the summertime GEM peak is relatively weaker at Zeppelin station than at Alert and Villum stations, likely due to the altitude (474 m a.s.l.) and relative remoteness of that station from the MIZ (Angot et al., 2016). These observations support the hypothesis that the MIZ is an important source of GEM in summer."

Q2: The authors mention that long-term observations from coastal stations may provide an incomplete understanding of the atmospheric Hg cycle in the central Arctic and lead to uncertainties in modeling atmospheric Hg in that region. This is important, how do the results from this study fill that gap?

Overall, I think this manuscript will contribute to the Hg research community and hopefully at a larger scale.

Re: Previous studies building on observations at coastal stations speculated that the summertime GEM maximum is mostly driven by oceanic Hg evasion from the Arctic Ocean (Angot et al., 2016; Hirdman et al., 2009). Based on the unique datasets collected in the central Arctic Ocean during the MOSAiC expedition, this study shows that oceanic evasion is not significant throughout the Arctic Ocean but mainly occurs in the Marginal Ice Zone. Moreover, current model simulations can accurately simulate the seasonal cycle of atmospheric Hg species, but do underestimate the amplitude of the seasonal variation in the Arctic (Dastoor et al., 2022a; Dastoor and Durnford, 2014). Our findings may thus help better parameterize key processes of Hg evasion in the Marginal Ice Zone and improve the overall model performance.

Q3: Finally, the big picture implications of the results are somewhat lacking. The manuscript would benefit from providing some discussion of the results in context of a changing climate, particularly with respect to a warming Arctic.

Re: Thank you for the great suggestion. We have added the following paragraph:

“Line 372: The Arctic is warming faster than the rest of the planet and Arctic sea-ice is declining rapidly with potential consequences for the Hg cycle (Yadav et al., 2020). Firstly, projected future Arctic warming might favor the conditions that stimulate AMDEs-driven Hg deposition through enhancing reactive halogen sources (Dastoor et al., 2022a), as previous reported long-term (1996-2017) dataset of BrO \cdot vertical column densities (VCDs) displayed an increasing trend of about 1.5 % of the tropospheric BrO \cdot VCDs per year during polar springs, which may be attributed to increase in first-year ice coverage that has a higher salinity than multiyear ice and facilitates the production of BrO \cdot (Bougoudis et al., 2020). This could enhance the deposition of Hg

and thus the load of Hg in surface seawater during the melt season. Secondly, previous study found that the width of MIZ in the warm season (July–September) in Arctic Ocean has increased by 13 km decade⁻¹ from 1979 to 2011, driven by the decrease of Arctic sea-ice extent and the replacement of thick, multi-year ice by thin, first-year ice (Strong and Rigor, 2013). In this context, it can be expected that the retreat of multi-year ice and the decline in sea-ice extent will continue, causing further increase in width and range of MIZ. This would translate into a larger source area of GEM in summertime. Thirdly, the decline in sea-ice extent and expanse in MIZ, can 1) transmit substantially more light into the underlying water column and make a longer growing season, leading to the increase in phytoplankton production (Arrigo et al., 2008; Pabi et al., 2008; Payne et al., 2022), and 2) promote the CO₂ uptake by open seawater and the biologic carbon pump process in MIZ (Bates et al., 2006; Qi et al., 2022), which would further increase the photo- and/or biological reduction capacity of Hg(II) here. As a result, we expect that in a context of Arctic warming, oceanic Hg evasion in the MIZ might become increasingly significant, and strengthen the role of the Arctic Ocean as a summertime Hg source. Given the relatively long atmospheric lifetime of GEM, this may have global consequences on the Hg cycle.”

Line by line comments:

Q4. Line 67 – two references here but only refers to the AMAP report

Re. Thanks. We have added another reference’s information (Chételat et al. (2022)) in the manuscript.

Q5. Line 73 – May want to provide some context, definition to AMDEs, since the first time introduced

Re. Thanks. We have added the corresponding information in the text, which was also showed below:

“**Line 77:** Since Schroeder et al. (1998) first reported the occurrence of springtime atmospheric Hg depletion events (AMDEs) in Alert (Nunavut, Canada), a phenomenon

during which GEM is near-quantatively oxidized to Hg(II) leading to Hg deposition onto snow/sea-ice, many long-term observation studies on atmospheric Hg have been conducted at several Arctic coastal monitoring stations (e.g., Alert, Zeppelin in Svalbard, and Villum Research Station in Northern Greenland) (Angot et al., 2016; Berg et al., 2013; MacSween et al., 2022; Skov et al., 2020).”

Q6: Line 78 – Word choice "unraveled"... revealed? Suggests this has been solved, yet the summertime maximum is still debated

Re. Thanks for your suggestion. We have revised the word “unraveled” as “revealed” based on your suggestion. The debate about the summertime maximum has been mentioned later (Line 83), so we would not put it here.

Q7: Line 85 – There are other important isotope studies to consider here, Douglas and Blum (2019); Zheng et al. (2021)

Re. Thanks. We have referred to and cited these two studies here, which is also shown below:

“**Line 96:** Hg re-emissions from the cryosphere could be fueled by AMDE deposited Hg that remains in the snowpack until the melt season. That fraction of Hg retained in the snowpack from spring to summer is, however, still highly uncertain. Using stable isotopes, Douglas and Blum (2019) found that 76 to 91% of AMDE deposited Hg is re-emitted prior to snowmelt in Alaskan snow. However, Zheng et al., (2021) estimated generally less photoreduction loss (0 to ~60%) from snow at Alert, with an average of $20 \pm 31\%$. These differences may be due to the complex factors that can affect the amount of Hg emitted from snowpack/sea-ice, such as solar radiation, Hg speciation in snowpack/sea-ice, halide and particulate matter concentrations of snowpack, snow temperature, snowfall frequency, and upward latent heat flux (Ferrari et al., 2005; Kamp et al., 2018; Mann et al., 2015).”

Q8. Line 89 – May be good to provide reference here to the coastal studies

Re. Thanks. We have cited the related references (Araujo BF et al., (2022)) here based on your comments.

Q9. Line 135 – define RD, might be worth indicating that these are used as a proxy for anthropogenic emissions

Re. Thanks. We have defined RD (solar radiation) in this section. We have also indicated CO and SO₂ as typical proxies for anthropogenic emissions in the revised manuscript based on your comments.

Q10. Line 139 – other studies could be referenced here as well

Re. Thanks. We have added the related reference (Wang et al. (2017)) here based on your comments.

Q11. Line 149 – check section reference (numbering is wrong?)

Re. Sorry for my editing error, and the section number here is “2.3.1”, not “3.3.1”. We have revised it.

Q12. Line 151 – this is interesting. Could authors provide reasoning as to why? Could it have occurred earlier in the year?

Re. Thanks. We think that one reason would be the air temperature. Low air temperature favors the production of reactive bromine radicals (Simpson et al., 2007) and the oxidation of GEM (by stabilizing the Hg(I) intermediate) (Steffen et al., 2008) and therefore favors the occurrence of AMDEs. AMDEs were actually observed in June at Zeppelin station when the temperature was between –5 and –10 °C, which were the lowest temperatures recorded for that month (Berg et al., 2013). The warmer air temperature (average: -0.86 ± 1.12 °C) in June during the MOSAiC expedition likely explains why no AMDEs were observed at this time. It should, however, be noted that AMDEs were ubiquitous during the MOSAiC springtime (March – May) as reported in Ahmed et al. (2023).

We have also added this discussion the revised manuscript (**Line 175**).

Q13. Line 156 - May be good place to note the significance of the summertime GEM maximum and knowing its origin

Re. Thanks for your suggestion. We have added the significance of the summertime GEM maximum here, which was also displayed below:

“**Line 187:** The summertime GEM maximum that follows springtime depletions (AMDEs) is a unique Arctic feature. **This Arctic GEM seasonality suggests a transition from a "mercury sink" in spring to a "mercury source" in summer (Araujo et al., 2022). Thus, understanding the origins and corresponding mechanisms of this GEM variability is of great importance for accurately assessing the regional mercury budget and associated ecological consequences.**”

Q14. Line 216 - additional references needed here

Re. Thanks. We have cited the corresponding references ((Galí et al., 2021); Park et al. (2018)) here based on your comments.

Q15. Line 231 - This is somewhat confusing because above the authors suggest AMDEs were not observed.

Re. In fact, AMDEs were not observed in summer during the MOSAiC expedition (June-September), which was the observation period of this study, but they were observed in spring of the MOSAiC expedition (March-May), which was reported in another study (Ahmed et al., 2023). We have clarified this in the revised manuscript.

Q16. Line 282 - somewhat vague statement. Also what about temperature on atmospheric chemistry (e.g., increasing temperature on emission of halogens/ oxidants, AMDEs, deposition of oxidized Hg to snow, regionally, MIZ as major sinks, etc.). Authors could draw on other studies here.

Re. Thanks for your suggestions. We have clarified this as follows:

“**Line 338:** Higher temperature could **not only** promote the release of Hg from the surface ocean and snowpack (Angot et al., 2016), **but also inhibit the oxidation process of GEM by reactive halogen radicals through causing the thermal decomposition of the Hg(I) intermediate (Steffen et al., 2008).**”

Q17. Line 285 to 289 – reference to help support explanation.

Re. Thanks. We have added the corresponding references ((Jin et al., 2021; Prijith et al., 2018)) here to support explanation.

Q18. Line 294 – it is not clear why WS outside of the range (at lower and higher ends of the range) would promote sea-air exchange and regional transport and why this specific range implies dilution. May need more clarity.

Re. Thanks. We have revised and more clearly clarified this content, which is also shown below:

“**Line 356:** Wind speed (WS) contributes 8% to GAM variance. **In stagnant meteorological condition characterized by low WS range (e.g. < 2.5 m/s in this study), the (re)-emitted Hg from the ocean might tend to accumulate, causing the positive correlation between GEM and WS (Zhang et al., 2021). As WS increases, this may enhance the atmospheric dilution of GEM, leading to the decreasing trend (Figure S6a)**”

Q19. Methods: What a sensitivity analysis completed on the model used? This can be added to supplemental information.

Re. Thanks. For sensitivity analysis of model, we used GAM to fit the non-linear relationship between the dependent variable (GEM) and independent variable (various influencing factors) to get a best fitting result of model, and subsequently evaluated the relative importance of these various influencing factors based on their respective F values (see section 2.3 in the manuscript). The bigger of F value, the more sensitivity

and the relative importance the corresponding factor will have. In addition, the factors with low F values and had insignificant contributions to the improvement of model fitting result (fitting R^2) will not be used. We have further clarified this selecting process of various factors which considered their sensitivities in model in section 3.4 of the manuscript, and added the information including F values of various factors to the supplementary material.

Q20. Figures: Some figures are fuzzy and difficult to read. Suggest higher resolution images (e.g., 4a, 4b, 5b)

Re. Thanks for your suggestions, and we have improved the qualities of corresponding images in the revised manuscript.

Reviewer #3:

General comments:

The study presents unique atmospheric GEM concentration data from the Arctic Ocean. The data were obtained using a Tekran 2537B system running on board of the research vessel Polarstern from June to September 2020. The study confirms an offshore summertime GEM peak between Greenland and Svalbard, a GEM maximum that has been observed at several coastal Arctic monitoring stations. The peak is not associated with atmospheric long-range transport of Hg. Instead a GAM simulation indicates that open ocean evasion of GEM, in particular from the low-latitude marginal ice zone between Greenland and Svalbard, is a significant source of GEM to the Arctic atmosphere in summer. While I want to point out the originality of the GEM dataset, I have some concerns regarding the identification of sources contributing to the GEM

summertime peak.

Q1. The authors report that 63.3% of the summertime GEM variation can be largely explained by open water re-emission and meteorological factors. Araujo et al., 2022, Nat. Commun. evidenced that the GEM peak originates from re-emission of Hg deposited to snow and sea ice during AMDEs. They concluded that GEM re-emission takes place directly from snow and sea ice but also from regional open waters that receive meltwater Hg inputs. How much of the GEM variation in the present study can be explained by direct GEM re-emission from snow and sea ice? Is re-emission from snow and sea ice not contributing to the GEM summertime peak in the central Arctic Ocean at all? Please include a paragraph in the ms about how direct snow and sea ice re-emission impacts GEM variation in summer.

Re. Indeed, GEM can be reemitted by both (1) snow and sea ice, and (2) oceanic evasion. We cannot quantify the contribution of direct GEM re-emission from snow and sea-ice due to the lack of relevant measurements and data in this study (e.g. Hg concentration and isotopic data in snow and sea-ice). This, however, does not change the main message of the manuscript that is re-emissions are mostly occurring in the MIZ, not throughout the Arctic Ocean. We speculate that the direct contribution of re-emission of Hg from snowpack/sea-ice to the summertime GEM peak may not be dominant, since the cruise observations of GEM and this study's PSCF analysis showed that the GEM concentrations and the regional contributions in the higher-latitude contiguous ice area are low (Figure 1 and Figure 5a in the manuscript). In addition, GEM measurements performed during a poleward transect early August during the MOSAiC expedition support this hypothesis, as GEM concentrations obviously decreased when *Polarstern* entered the pack ice with increased distance from the MIZ (please see the response of your Q3). More studies are needed to further evaluate the role of direct Hg re-emission from snowpack and sea-ice in GEM variation during Arctic summertime. We have also added some of these contents to **Line 314**

Q2. Long-range transport of Hg contributes only marginally to the GEM summertime peak. Meteorological factors, however, explain 37% of the identified GEM variation in the central Arctic Ocean. Advection of Hg from the mid-latitudes (i.e. long-range transport of anthropogenic emissions and terrestrial Hg) and Hg evading from the ocean can indeed be considered sources contributing to GEM levels in the Arctic atmosphere. However, please explain how meteorological factors can be a source of GEM. Meteorological factors such as wind speed or solar irradiation have been demonstrated to influence the rate of GEM evasion from the ocean but cannot be considered a source in my understanding.

Re. We agree with you that meteorological factors cannot be classified as a source but are a potentially important influencing factor of GEM in the Arctic Ocean. As the investigation of the source of GEM in the Arctic Ocean is the focus of this paper, we emphasize it in the abstract and introduction. We also revised the original subtitle: “2.3.3 Meteorological factors” of section 2.3 as an independent title: “2.4 Impact of meteorological factors”.

Q3. In July 2020, when the GEM peak was detected, the Polarstern was cruising between Greenland and Svalbard, correct? I wonder whether the July GEM peak could have been detected close to the North Pole as well ? Did you find any indication for that? Please add dates and the respective location of the Polarstern in Fig. 1a.

Re. Yes. When the summertime GEM peak was detected, the Polarstern was cruising between Greenland and Svalbard. Due to the geographical differences, we do not know whether the July GEM peak could have been observed close to the North Pole. In early August, however, the ship transited northward from the MIZ to a new ice floe area located close to the North Pole, and a clear decline of GEM concentrations was observed, with GEM decreasing to background concentrations when entering the pack ice (Figure 1). Based on these results, we strongly believe that oceanic Hg evasion would have been low close to the North Pole in July as well.

Figure 1. The Latitudinal and the corresponding time series of GEM concentrations during the ship transited northward from the MIZ to the North Pole.

To more clearly show the corresponding information of locations and dates of the *Polarstern*, we have added their information in Figure S3 based on your comments, which is also shown below:

Figure 2. Spatial distribution of observed gaseous elemental mercury (GEM) measurements in the Arctic Ocean during the summer legs (June–September) of the MOSAiC expedition. The grey boxes mark the corresponding locations and date during the expedition

Q4. Overall, ocean evasion explains less than 50% of the GEM variation in summer (L404: 36.7 % of the GEM variation cannot be explained). Still, there is strong indication for large GEM evasion from the MIZ between Greenland and Svalbard. It would strengthen the manuscript, if the authors could include a statement about the GEM flux rate ($\text{ng m}^{-2} \text{ day}^{-1}$) that could be expected from the MIZ between Greenland and Svalbard. What flux rate would be required to increase the GEM concentration within the Arctic Ocean boundary layer from 1.35 ng m^{-3} (September, background) to

1.8 ng m⁻³ (July, summertime peak). Under the assumption that GEM re-emission from the other parts of the central Arctic Ocean remain small.

Re. Thanks for your comments. To estimate the flux rate of GEM from the MIZ between Greenland and Svalbard that is necessary to explain the increase in GEM concentration within the Arctic Ocean boundary layer from 1.3 ng/m³ to 1.8 ng/m³, the following assumptions are made here:

(1) Neglect other sources of GEM in the Arctic Ocean and the GEM exchange between the Arctic Ocean and other areas;

(2) The GEM emitted from the MIZ will affect its nearby Arctic Ocean area that is north of 70°N, as the summertime GEM maximum phenomenon was mainly observed at high-latitude (north of 70°N) Arctic stations (e.g. Zeppelin, Alert, Villum stations), while low-latitude Arctic stations (e.g. Andøya) observed no apparent GEM maximum phenomenon (Angot et al., 2016). In this study, the speculated range of the Arctic Ocean area that the GEM evasion from the MIZ between Greenland and Svalbard would influence is shown below (Figure 3, the fan-shaped area with red border is the range of influence, and the triangle area with yellow border is the MIZ that serves as the GEM source):

Figure 3. The speculated range of Arctic Ocean area that GEM evasion from the MIZ between Greenland and Svalbard would influence. The fan-shaped area with red border is the range of influence, and the triangle area with yellow border is the MIZ that is served as GEM source)

By geometric calculation, the area of MIZ that emits GEM accounts for about 6.25% of its range of influence in the Arctic Ocean.

(3) Based on the time series of GEM in this study, in which the GEM concentration showed apparent increases during three periods in July (from about 2020/6/19 to 2020/7/2, from 2020/7/4 to 2020/7/9, and from 2020/7/11 to 2020/7/18, Figure 4) and mainly occurred in the MIZ, we set the duration of GEM evasion in the MIZ as 20 days.

Figure 4. The time series of GEM during the summer legs (June–September) of the MOSAiC expedition, in which the period of summertime GEM peaks in July is marked.

Based on the assumptions above, the GEM flux rate can be calculated using the following equation, according to the principle that mercury evaded from the MIZ in summer leads to the increase in observed GEM concentration in its range of influence in the Arctic Ocean:

$$\text{Flux} \times 0.0625S \times 20 = (1.80 - 1.30) \times S \times \text{PBLH}$$

Where Flux is GEM evasion flux rate ($\text{ng}\cdot\text{m}^{-2}\cdot\text{day}^{-1}$) from the MIZ between Greenland and Svalbard; S is the range of influence (km^2) in the Arctic Ocean by GEM evasion from the MIZ; 1.80 (ng/m^3) and 1.30 (ng/m^3) are July peak concentration and September background concentration of GEM in the Arctic respectively; 0.0625 is the area fraction of the MIZ to its range of influence in the Arctic Ocean; and PBLH is the planetary boundary layer height (m) of the Arctic Ocean. Here we set this value as the July average (140m) during the MOSAiC cruise. The calculated flux is $56 \text{ ng}\cdot\text{m}^{-2}\cdot\text{day}^{-1}$

¹, which is more than twice the flux in the Arctic ice-free open ocean ($< 24 \text{ ng}\cdot\text{m}^{-2}\cdot\text{day}^{-1}$) (Andersson et al., 2008; DiMento et al., 2019), while lower than that observed in coastal areas in the Canadian Arctic Archipelago ($\sim 130 \text{ ng}\cdot\text{m}^{-2}\cdot\text{day}^{-1}$) (Kirk et al., 2008). We also added this estimation to Section 2.3.2, which is also shown below:

“Line 321: Overall, we suggest that oceanic evasion in the MIZ combined with regional transport is the dominant GEM source in the central Arctic – a phenomenon that likely explains the observed summertime GEM maximum. A back-of-the-envelope calculation (see Text S1) suggests a **Hg evasion flux of $56 \text{ ng}\cdot\text{m}^{-2}\cdot\text{day}^{-1}$ in the MIZ. This flux is more than twice the flux measured in the Arctic open ocean ($< 24 \text{ ng}\cdot\text{m}^{-2}\cdot\text{day}^{-1}$) (Andersson et al., 2008; DiMento et al., 2019), but lower than that measured in coastal areas of the Canadian Arctic Archipelago ($\sim 130 \text{ ng}\cdot\text{m}^{-2}\cdot\text{day}^{-1}$) (Kirk et al., 2008). It is also an order of magnitude higher than the Arctic Ocean evasion flux reported by Dastoor et al. (2022) in their latest Arctic Hg mass balance budget (23–45 tons/year, i.e., $3.7\text{--}7.3 \text{ ng}\cdot\text{m}^{-2}\cdot\text{day}^{-1}$ assuming a surface area of $1.7 \times 10^7 \text{ km}^2$). Further dedicated field and modeling studies are needed to further constrain that flux.”**

Q5. Such a statement could be integrated into a yet missing “conclusion” or “implication” paragraph. I strongly suggest to add conclusions and implications at the end of the main text. Please discuss the significance of your results and how your findings fit with our current understanding of the Arctic mercury cycle (cp. with Dastoor et al., 2022, Nature reviews, earth and environment and Araujo et al., 2022, Nat. Commun.).

Re. Thanks for your important comments. We have added the corresponding conclusions and implications in which we discuss how our findings fit with our current understanding of the Arctic mercury cycle and highlight the significance of our results.

We also added the discussion about how our findings will change with respect to a warming Arctic, which are also shown below:

“Line 363: Based on the unique whole summertime observation of GEM in the central Arctic Ocean during the MOSAiC expedition, we show that oceanic evasion is an important source of atmospheric Hg in summertime.. This study further verifies our current understanding of the Arctic Hg cycle, specifically regarding the source of the summertime GEM maximum and the general role of the Arctic Ocean as a Hg source during the summertime (Araujo et al., 2022; Dastoor et al., 2022a). Furthermore, this study offers new insights by showing that oceanic evasion is not significant throughout the Arctic Ocean but mainly occurs in the MIZ. Our estimate of the Hg evasion flux suggests a higher magnitude than in the open ocean, causing a rapid increase of GEM concentrations throughout the Arctic.

The Arctic is warming faster than the rest of the planet and Arctic sea-ice is declining rapidly with potential consequences for the Hg cycle (Yadav et al., 2020). Firstly, projected future Arctic warming might favor the conditions that stimulate AMDEs-driven Hg deposition through enhancing reactive halogen sources (Dastoor et al., 2022a), as previous reported long-term (1996-2017) dataset of BrO \cdot vertical column densities (VCDs) displayed an increasing trend of about 1.5 % of the tropospheric BrO \cdot VCDs per year during polar springs, which may be attributed to increase in first-year ice coverage that has a higher salinity than multiyear ice and facilitates the production of BrO \cdot (Bougoudis et al., 2020). This could enhance the deposition of Hg and thus the load of Hg in surface seawater during the melt season. Secondly, previous study found that the width of MIZ in the warm season (July–September) in Arctic Ocean has increased by 13 km decade⁻¹ from 1979 to 2011, driven by the decrease of Arctic sea-ice extent and the replacement of thick, multi-year ice by thin, first-year ice (Strong and Rigor, 2013). In this context, it can be expected that the retreat of multi-year ice and the decline in sea-ice extent will continue, causing further increase in width and range of MIZ. This would translate into a larger source area of GEM in summertime. Thirdly, the decline in sea-ice extent and expanse in MIZ, can 1) transmit substantially more light into the underlying water column and make a longer growing season, leading

to the increase in phytoplankton production (Arrigo et al., 2008; Pabi et al., 2008; Payne et al., 2022), and 2) promote the CO₂ uptake by open seawater and the biologic carbon pump process in MIZ (Bates et al., 2006; Qi et al., 2022), which would further increase the photo- and/or biological reduction capacity of Hg(II) here. As a result, we expect that in a context of Arctic warming, oceanic Hg evasion in the MIZ might become increasingly significant, and strengthen the role of the Arctic Ocean as a summertime Hg source. Given the relatively long atmospheric lifetime of GEM, this may have global consequences on the Hg cycle.”

Specific comments:

Q6. Line 62: There are more recent studies reporting atmospheric lifetimes of GEM. Consider referring to Saiz-Lopez et al., 2018, Nat. Commun. or Horowitz et al., 2017, ACP.

Re. Thanks for your suggestion. We have referred to these references and cited them based on your comment.

Q7. Line 67: The AMAP report is referenced with number 44 instead of 8. Is there a reason for that? Adjust reference numbering throughout the ms.

Re. Thanks for your suggestion. I have not concerned the order of references we cited before I have revised it and checked the corresponding reference numbering throughout the manuscript.

Q8. Line 90: Delete “unfortunately” .

Re. Thanks. We have deleted it.

Q9. Line 135: Write out “RD” . You define all abbreviations in the Method section. Please do that in main text instead.

Re. Thanks for your comment. We have added the corresponding detailed definitions of the abbreviations in main text based on your comment.

Q10. Line 194: “Central Arctic Ocean” or “central Arctic Ocean”? Be consistent.

Re. Thanks. We have consisted this term as “central Arctic Ocean” throughout the manuscript.

Q11. Line 218: What would be an important factor? This sentence is hard to understand. Please rephrase.

Re. Thanks. We have rephrased this sentence, which is also shown below: “**These phenomena indicated that oceanic evasion of Hg and its regional transport from the MIZ would be an important factor that drives the summertime GEM maximum in the Arctic Ocean, as we observed in this study.**”

Q12. Line 236: Write out “Chla” .

Re. Thanks. We have written out “Chla” (chlorophyll-a) in the revised manuscript.

Q13. Line 270: Write out as follows: Wind speed, atmospheric pressure and air temperature. Stick to “re-emission” .

Re. Thanks for your suggestion. We have revised them based on your comments, which are also shown below:

“**Line 361:** Meteorological conditions (**wind speed, atmospheric pressure and air temperature**) can influence the **re-emission**, transport and dilution process of atmospheric Hg.”

Q14. L313: The flow rate of the Tekran 2537B was 0.7 L min⁻¹ which is low compared to most Tekran deployments. Is there a reason why the authors used such a low flow rate? Analysis of GEM in 3.5 L of air is at the lower limit. Please explain briefly why an additional Tekran analyzer was setup even though GEM was determined

in the CU container on a 15 min resolution? Please also add a comment about how you avoided to measure GEM that potentially originated from ship exhaust gas emissions.

Re. During summertime observations of GEM, two 0.45 μm Teflon filter clips were installed to prevent oceanic sea-salt aerosols from entering the Tekran 2537X/BTM instrument system. Considering that this may increase the air resistance of the sampling system during the measurement, we set the flow rate lower than 1.0 L/min. In fact, the injection flow of the Tekran 2537X/BTM instrument is normal in the range of 0.5 L/min ~1.5 L/min, and will not affect the GEM measurement during Arctic summertime in this study. The sampling inlet was mounted at the front of the ship to minimize the influence of the ship exhaust and data were carefully screened for local contamination as discussed in Angot et al. (2022) and Beck et al. (2022). The comparisons between the measured GEM and the typical exhaust components (including SO₂, particulate and CH₄) show that there was minimal influence of ship exhaust on our GEM measurements (Figure 5). We have also added this content to method section in the manuscript. Please also note that it is quite common to have redundant measurements during such an expedition to limit gaps in time-series and allow cross-evaluation. As discussed in Angot et al. (2022), redundant measurements were for example performed for carbon dioxide, methane, carbon monoxide, nitrous oxide, ozone or volatile organic compounds.

Figure 5. Time series of GEM and the typical exhaust components (including SO₂, particulate (Num. Conc.) and CH₄) during both the whole observation period and the high concentration periods

(marked by red box). It displays that the significant increases of exhaust components correspond to no apparent increase of GEM meanwhile (e.g. grey shaded area)

Q15. L 685, 697, 709: Add units to the axis text in Figs. 3a, b, 4a, b, 5b.

Re. Thanks, we have added units to the axis text based on your comments.

Supporting information:

Q16. L 38: Is that a table of content? Please adjust.

Re. Thanks. We have adjusted it in form of a table.

Q17. L 45: Why is the section title number 3.2? Please adjust.

Re. Sorry of this editing error, this section title is redundant and we have removed it.

Q18. L 70: How was this figure created? Using HYSPLIT? Information is missing in the caption.

Re. Thanks. Yes, this figure was created using the HYSPLIT transport and dispersion model from NOAA Air Resources Laboratory (ARL). We have also added this information in the caption.

Q19. L 95: Add units to the axis text in Fig S7.

Re. Thanks. We have added them in the revised Figure S7.

Reviewer #4:

This study investigated key mechanisms for occurrence of the summer concentration

peak of gaseous elemental mercury (GEM) in the Arctic based on observations during the Multidisciplinary drifting Observatory for the Study of Arctic Climate (MOSAiC) expedition and a statistical modeling approach. Oceanic evasion in the Marginal Ice Zone was found to be the dominance source for the summertime GEM maximum, and the photoreduction of Hg(II) to GEM by phytoplankton was proposed to be the main mechanism. The observational data obtained in the study are valuable, and the finding reported in the study is interesting which could draw widespread attention.

Q1. However, the generalized additive model (GAM) used in this study needs more information on model validation, and more solid evidence needs to be provided for the mechanisms the authors proposed. A comprehensive schematic diagram for what's going on during the summer GEM maximum is encouraged to be added to the manuscript for readers to get a whole picture of the crucial mechanisms. Overall in my opinion, substantial improvement is required before the manuscript is acceptable for publication on Nature Communications.

Re. Thanks for your comments. For the validation of GAM method, we have substantially referred to previous studies about environmental areas that used the GAM method (Gong et al., 2017; Shi et al., 2022; Wu et al., 2020; Zhang et al., 2021; Zhao et al., 2022) and added section 3.5 and added more information in discussing the model validation, which is also shown below:

“3.5 Model validation

We further systematically evaluated the quality of the GAM using different methods. We used a 5-fold cross-validation test to assess the accuracy of GAM simulation. The principle of this method is randomly dividing the whole dataset into 5 subsets, and in each cross-validation round, four subsets are used to fit the model, and the remaining subset is used to predict. This process was repeated 5 times to ensure that every subset was tested (Shi et al., 2022; Wu et al., 2020; Zhang et al., 2021). The 5-fold cross-validation results displayed a good coincidence between the GAM and cross-validated results (slope=1.00, $R^2=0.99$), demonstrating good reliability of the model (Fushiki, 2011)(Figure 3b). In order to test the underlying assumptions of homogeneity, normality,

and independence of GAMs to ensure the validity and accuracy of the model, we used the following methods: (1) quantile-quantile (Q-Q) plot (sample quantiles against theoretical quantiles), (2) scatterplots of residuals against linear predictors, and (3) histograms of the residuals (Zhang et al., 2021). The Q-Q plot result showed that GAM produced good results around the average concentration; the scatterplot of residuals vs. linear predictor displayed that residuals were generally concentrated around 0 value, and presented a random distribution with no obvious trend, indicating unbiased simulations of GEM. The histogram of residuals close to normal distribution, which suggests that the error of model fitting is random and that the selection of predictor variables is reasonable (Figure S1). These results suggest that the selected predictors will be useful for identifying sources of Hg in the central Arctic atmosphere.”

We have also added a comprehensive schematic diagram for what’s going on during the summer GEM maximum based on your comments, which is also shown below:

Figure 6. Diagram of the main processes that drive the significant Hg(0) (or GEM) (re)-emission in Marginal Ice Zone, including (1) previously deposited Hg(II) during springtime AMDEs is transferred to the surface ocean with melting ice/snow-water, resulting in a large Hg load in surface seawater in the MIZ; (2) high phytoplankton mass in MIZ may lead to a high reduction capacity for Hg(II) in MIZ, as the excretion of photoreactive organic compounds by phytoplankton can facilitate the photoreduction of Hg(II) to Hg(0) via electron transfer. In addition, phytoplankton can also directly contribute to the reduction of Hg(II) to Hg(0) through an enzymatic detoxification mechanism linked to their photosynthetic activity; and (3) the melting of sea-ice and the absence of upper ocean stratification in the MIZ (i.e., absence of a meltwater layer) facilitates air–sea gas exchange.

Here are some specific comments:

Q2. Abstract: The most probable cause of the summer GEM peak, the photoreduction of Hg(II) to GEM by phytoplankton (if I understand the Results and Discussion part correctly), is not reflected in the abstract. The authors only mentioned that oceanic evasion mainly occurs in the Marginal Ice Zone, which could mislead the readers to think that the air – ice Hg exchange might be the dominant mechanism of the summer GEM maximum.

Re. Thanks for your comments. Yes, the photoreduction of Hg(II) to GEM facilitated by high phytoplankton mass is likely to be an important contributor to GEM evasion in the MIZ. Apart from that, the high Hg input from melting ice/snowpack in MIZ, and the loss of sea-ice as a barrier of Hg sea-air exchange in the MIZ may also be important. We have briefly added these factors in the abstract, which is also shown below:

“**Abstract:** Atmospheric gaseous elemental mercury (GEM) concentrations in the Arctic exhibit a clear summertime maximum, while the origin of this peak is still a matter of debate in the community. Based on summertime observations during the Multidisciplinary drifting Observatory for the Study of Arctic Climate (MOSAIC) expedition and a modeling approach, we further investigate the sources of Arctic atmospheric Hg. **Simulations with a generalized additive model (GAM)** show that long-

range transport of anthropogenic and terrestrial Hg from lower latitudes is a minor contribution (~2%). The dominant source (> 50%) is oceanic evasion. **A potential source contribution function (PSCF) analysis** further shows that oceanic evasion is not significant throughout the Arctic Ocean but mainly occurs in the Marginal Ice Zone (MIZ) **due to the specific environmental conditions in that region. Our results suggest that this regional process could be the leading contributor to the observed summertime GEM maximum. In the context of rapid Arctic warming and the observed increase in width of the MIZ, oceanic Hg evasion may become more significant and strengthen the role of the Arctic Ocean as a summertime source of atmospheric Hg.”**

Q3. Line 82: The citation should be Durnford et al. according to the reference list.

Re. Sorry for my editing error. Yes, the citation should be Durnford et al and we have revised it.

Q4. Lines 82 – 87: The summary of the GEM source apportionment results could cause misinterpretation. Long-range transport of Asian air is a primary source. Legacy Hg emissions or re-emissions from terrestrial regions or ice zones are considered as secondary sources. Due to the different methodologies adopted, a certain part of the discrepancy between these studies could be caused by the categorization of GEM sources. Some legacy Hg from historical Asian air could be accumulated in plants, snowpack and ice in the Arctic area and reemitted as a secondary source of GEM. It would be helpful for the readers to understand the Arctic Hg cycle if the authors could make a more comprehensive introduction here, incorporating these literatures.

Re. Thanks for your important comments. We have added more information about the Hg cycle process in the Arctic region and revised the corresponding statements in this section, which is also shown below:

“Line 72: Long-range transport of air, riverine input and coastal erosion from circumpolar land can be primary Hg sources to Arctic area (Dastoor et al., 2022b; Fisher et al., 2012). A fraction of legacy Hg from these historical primary sources can accumulate in seawater, plants, glaciers and ice sheets in the Arctic and later be re-

emitted as secondary emissions of atmospheric Hg (Douglas and Blum, 2019; Hirdman et al., 2009; Zheng et al., 2021).

Line 88: A modeling study by Durnford et al. (2010) suggested that long-range transport of Asian air is the most important primary source of atmospheric Hg to the Arctic in all seasons. Sommar et al. (2010) and Fisher et al. (2012) later attributed enhanced summertime GEM concentrations to ocean Hg evasion in the Arctic Ocean fed by terrestrial Hg inputs (rivers and coastal erosion) based on observations and several sensitivity runs using GEOS-Chem model. Recent isotopic work suggests, however, the dominant role of re-emissions from the Arctic cryosphere, while the role of terrestrial Hg from rivers and coastal erosion was found to be minor (Araujo et al., 2022).”

Q5. Line 116: Misspelling of “coastal” .

Re. Sorry for my editing error, and we have revised it.

Q6. Section 2.1: The discussion in this section is limited. Significance of difference from previous studies based on t-test should be given. The standard deviation of the central Arctic GEM observations during August 25 – September 8, 2012 (0.61) is much higher than that in this study (0.07). What caused the large fluctuation in the study of Yu et al. (2014)?

Re. Thanks. We have added the t-test information to the manuscript. For the difference in the standard deviation of data between this study and Yu et al., (2014), we think that an important reason might be the large differences in their corresponding observation sites and the corresponding sea-ice cover. In fact, the research area that is defined as “central Arctic Ocean” in Yu et al., (2014) has a large difference from this study (Yu et al., (2014): 80°N–87.6°N, 9.2°E–168.9°W; this study: 87.80°N–88.75°N, 104.27°E–120.83°E), which can cause a large difference in sea-ice conditions (see Figure 7). The dense sea-ice cover (average sea-ice fraction: 0.92±0.07) in this study would largely inhibit the air-sea exchange of Hg, which may cause the apparently more stable variation of GEM concentration. The oceanic evasion of Hg would be generally

more significant in the research area in Yu et al. (2014), in which the sea-ice fraction was significantly lower (0.36 ± 0.32) (Figure 7), and may lead to larger fluctuations of GEM compared with this study. We have clarified the corresponding statements as:

“**Line 137:** In addition, our observed GEM average concentration ($1.32 \pm 0.07 \text{ ng/m}^3$) are comparable to the reported GEM observation ($1.40 \pm 0.61 \text{ ng/m}^3$; t-test: $p < 0.01$) during the same period (August 25–September 4) in 2012 in the central Arctic Ocean (80°N – 87.6°N , 9.2°E – 168.9°W), while with apparently lower standard deviation than the latter (Yu et al., 2014). This difference in standard deviation may to some extent be attributed to the significantly higher and less variable sea-ice fraction (0.92 ± 0.07) in this study, which inhibited the oceanic evasion of Hg and led to the apparently more stable variation of GEM concentration, compared with Yu et al., (2014) (with average sea-ice fraction of 0.36 ± 0.32).”

Figure 7. The spatial distribution of sea-ice during the same period (August 25–September 4) in Yu et al., (2014) and in this study.

Q7. Lines 149 – 151: What could be the reason for the absence of this phenomenon in the MOSAiC expedition (especially in June)? Why did the authors present this comparison here?

Re. In fact, AMDEs were just not observed in summer during the MOSAiC expedition (June-September), which was the observation period of this study, but they were observed in spring (March-May), as was reported in another study (Ahmed et al., 2023). We present this comparison here to further illustrate the conditions of the occurrence of AMDEs.

We think that one reason would be the air temperature. Low air temperature favors the production of reactive bromine radicals (Simpson et al., 2007) and the oxidation of GEM (by stabilizing the Hg(I) intermediate) (Steffen et al., 2008) and therefore favors the occurrence of AMDEs. AMDEs were actually observed in June at Zeppelin station when the temperature was between -5 and -10 °C, which were the lowest temperatures recorded for that month (Berg et al., 2013). The warmer air temperature (average: -0.86 ± 1.12 °C) in June during the MOSAiC expedition likely explains why no AMDEs were observed at this time. It should, however, be noted that AMDEs were ubiquitous during the MOSAiC springtime (March – May) as reported in Ahmed et al. (2023). We have also added this discussion the revised manuscript (**Line 175**).

Q8. Figure 4c: What is the significance of the relationship between land fraction and GEM? The negative correlation seems not quite robust to me.

Re. In this section we mainly investigated the variation of GEM average concentrations with the increase of the corresponding land fraction, aimed at understanding whether the higher 168 h backward trajectory's transport time over land compared to the total transport time (land fraction) can cause the higher level of GEM. The significance of their correlation was not our focus here. Based on your comments, we explored the correlation between land fraction and average GEM here, and found their negative linear correlation (Figure 8). Although this linear correlation is insignificant ($p=0.056$), it at least indicates that the average concentrations of GEM do not increase with increased land fraction. Additionally, this analysis to some extent shows the minor contribution of terrestrial anthropogenic Hg emissions to GEM in the Arctic Ocean during the summertime. We have also replaced the original Figure 4c with

the figure shown below, and revised the corresponding statement in the manuscript, which is also shown below:

“**Line 214:** We find that **the average GEM concentration slightly decreases as the land fraction increases** (Figure 4c), **which to some extent suggests a** minor contribution of **land-based** Hg emissions to GEM in the Arctic Ocean during the summertime.”

Figure 8. Relationship between the fraction of the 168 h backward trajectory’s transport time over land area over the total transport time (Land fraction (Traj)) and its corresponding average GEM concentration

Q9. Lines 182 – 194: The reasoning is a bit farfetched. More solid evidence should be provided to support the speculation.

Re. The seasonality of poleward pollution transport from mid-latitudes and the influence of the Arctic dome is well documented in the literature (Boyer et al., 2023; Bozem et al., 2019; Klonecki et al., 2003; Law and Stohl, 2007; Willis et al., 2018). A previous model study that reported the simulation of atmospheric Hg global spatial distribution also suggested that advection of mercury from mid-latitudes to the Arctic

is insignificant in summer due to weak airflow and to a confined polar front (Dastoor and Larocque, 2004). This is also reflected in the 7-day backward trajectories during the whole observation period (June-September) in this study, which show that air masses were mainly concentrated over the Arctic Ocean, while the air masses from the circumpolar land area were generally scarce (Figure S2 in the supporting information of the manuscript).

Q10. Figure 5b: What do these three peaks stand for in this influencing pattern? This is an important question to verify the reliability of the GAM model.

Re. Thanks for your comments. For the reliability of the GAM model, we have systematically evaluated it using a series of widely used methods for the GAM model, demonstrating good reliability of the model without overfitting (please see the newly added section 3.5). In fact, these three peaks in the GAM simulation basically reflect the complex sea-ice conditions in the lower-latitude marginal ice zone (MIZ), in which the summertime GEM peaks characterized by elevated concentration levels and high fluctuations were also observed. This can be suggested by the observed relationship between sea-ice and GEM (Figure 9), which shows that the most apparent GEM fluctuations occur at sea-ice fractions of approximately 0.5 and 0.8, in accordance with the two GEM peaks that occur at Open-water fractions of approximately 0.5 and 0.2 in Figure 5b, and the corresponding low latitudes (Figure 9) during GEM peaks indicate the MIZ area. The spatial distributions of GEM and sea-ice also present these features, showing that the GEM peaks occurred in the MIZ with complex sea-ice conditions (Figure 10).

Figure 9. The relationship between GEM, sea-ice fraction and the corresponding latitude during the whole observation period.

Figure 10. The locations of summertime GEM peaks during the MOSAiC expedition.

Q11. Line 230: How much Hg(II) is stored yearly approximately? More quantitative comparisons are required.

Re. Thanks. Dastoor et al. (2022) suggested that approximately 9 tons (4 ~ 15 tons) of Hg are stored in the Arctic Ocean sea-ice with sea-ice melt contributing 1.4 ± 0.4 tons of Hg to the Arctic Ocean. We have also added this discussion in **Line 281**.

Q12. Line 239: It should be Figure 7.

Re. Sorry for my editing error, and we have revised it.

Q13. Section 2.3.3: Solar radiation should be included as a meteorological predictor in the GAM model. In the previous section, the photoreduction of Hg(II) to GEM by phytoplankton is proposed to be a dominant mechanism. If so, solar radiation is supposed to be a key factor for the summertime GEM maximum.

Re. Thanks for your comments. We found that in the GAM model, the contribution of solar radiation to GEM was minor (1.3%), with respectively lower p-value (0.00049) compared with other parameters. We speculated that the low contribution of solar radiation in the GAM model would be to some extent due to two factors: First, the diurnal variation of hourly solar radiation data can influence the statistical results in GAM. As we used daily data of solar radiation instead to avoid the influence of its diurnal variation, we found that solar radiation displayed moderate positive linear correlation with GEM (Figure 11), in accordance with concurrent elevated solar radiation during the GEM and Chla peaks (Figure 6 in the manuscript), which may support the photoreduction of Hg(II) to GEM. The second factor is the attenuation of solar radiation caused by marine phytoplankton and sea-ice in the MIZ.

Figure 11. The relationship between daily GEM and solar radiation.

References

- Ahmed, S., Thomas, J.L., Angot, H., Dommergue, A., Archer, S.D., Bariteau, L., Beck, I., Benavent, N., Blechschmidt, A.-M., Blomquist, B., Boyer, M., Christensen, J.H., Dahlke, S., Dastoor, A., Helmig, D., Howard, D., Jacobi, H.-W., Jokinen, T., Lapere, R., Laurila, T., Quéléver, L.L.J., Richter, A., Ryjkov, A., Mahajan, A.S., Marelle, L., Pfaffhuber, K.A., Posman, K., Rinke, A., Saiz-Lopez, A., Schmale, J., Skov, H., Steffen, A., Stupple, G., Stutz, J., Travnikov, O., Zilker, B., 2023. Modelling the coupled mercury-halogen-ozone cycle in the central Arctic during spring. *Elementa: Science of the Anthropocene* 11, 00129. <https://doi.org/10.1525/elementa.2022.00129>
- Andersson, M.E., Sommar, J., Gårdfeldt, K., Lindqvist, O., 2008. Enhanced concentrations of dissolved gaseous mercury in the surface waters of the Arctic Ocean. *Marine Chemistry* 110, 190-194. <https://doi.org/10.1016/j.marchem.2008.04.002>
- Angot, H., Blomquist, B., Howard, D., Archer, S., Bariteau, L., Beck, I., Boyer, M., Crotwell, M., Helmig, D., Hueber, J., Jacobi, H.-W., Jokinen, T., Kulmala, M., Lan, X., Laurila, T., Madronich, M., Neff, D., Petäjä, T., Posman, K., Quéléver, L., Shupe, M.D., Vimont, I., Schmale, J., 2022. Year-round trace gas measurements in the central Arctic during the MOSAiC expedition. *Scientific Data* 9, 723. <https://doi.org/10.1038/s41597-022-01769-6>
- Angot, H., Dastoor, A., De Simone, F., Gårdfeldt, K., Gencarelli, C.N., Hedgecock, I.M., Langer, S., Magand, O., Mastro Monaco, M.N., Nordstrøm, C., Pfaffhuber, K.A., Pirrone, N., Ryjkov, A., Selin, N.E., Skov, H., Song, S., Sprovieri, F., Steffen, A., Toyota, K., Travnikov, O., Yang, X., Dommergue, A., 2016. Chemical cycling and deposition of atmospheric mercury in polar regions: review of recent measurements and comparison with models. *Atmos. Chem. Phys.* 16, 10735-10763. <https://doi.org/10.5194/acp-16-10735-2016>
- Araujo, B.F., Osterwalder, S., Szponar, N., Lee, D., Petrova, M.V., Pernov, J.B., Ahmed, S., Heimbürger-Boavida, L.-E., Laffont, L., Teisserenc, R., Tananaev, N., Nordstrom, C., Magand, O., Stupple, G., Skov, H., Steffen, A., Bergquist, B., Pfaffhuber, K.A., Thomas, J.L., Scheper, S., Petäjä, T., Dommergue, A., Sonke, J.E., 2022. Mercury isotope evidence for Arctic summertime re-emission of mercury from the cryosphere. *Nature Communications* 13, 4956. <https://doi.org/10.1038/s41467-022-32440-8>
- Ariya, P.A., Amyot, M., Dastoor, A., Deeds, D., Feinberg, A., Kos, G., Poulain, A., Ryjkov, A., Semeniuk, K., Subir, M., Toyota, K., 2015. Mercury Physicochemical and Biogeochemical Transformation in the Atmosphere and at Atmospheric Interfaces: A Review and Future Directions. *Chemical Reviews* 115, 3760-3802. <https://doi.org/10.1021/cr500667e>
- Arrigo, K.R., van Dijken, G., Pabi, S., 2008. Impact of a shrinking Arctic ice cover on marine primary production. *Geophysical Research Letters* 35. <https://doi.org/10.1029/2008GL035028>
- Bates, N.R., Moran, S.B., Hansell, D.A., Mathis, J.T., 2006. An increasing CO₂ sink in the Arctic Ocean due to sea-ice loss. *Geophysical Research Letters* 33. <https://doi.org/10.1029/2006GL027028>
- Beck, I., Angot, H., Baccharini, A., Dada, L., Quéléver, L., Jokinen, T., Laurila, T., Lampimäki, M., Bukowiecki, N., Boyer, M., Gong, X., Gysel-Beer, M., Petäjä, T., Wang, J., Schmale, J., 2022. Automated identification of local contamination in remote atmospheric composition time series.

- Atmos. Meas. Tech. 15, 4195-4224. <https://doi.org/10.5194/amt-15-4195-2022>
- Ben-Bassat, D., Shelef, G., Gruner, N., Shuval, H.I., 1972. Growth of Chlamydomonas in a Medium containing Mercury. *Nature* 240, 43-44. <https://doi.org/10.1038/240043a0>
- Berg, T., Pfaffhuber, K.A., Cole, A.S., Engelsen, O., Steffen, A., 2013. Ten-year trends in atmospheric mercury concentrations, meteorological effects and climate variables at Zeppelin, Ny-Ålesund. *Atmos. Chem. Phys.* 13, 6575-6586. <https://doi.org/10.5194/acp-13-6575-2013>
- Bougoudis, I., Blechschmidt, A.M., Richter, A., Seo, S., Burrows, J.P., Theys, N., Rinke, A., 2020. Long-term time series of Arctic tropospheric BrO derived from UV-VIS satellite remote sensing and its relation to first-year sea ice. *Atmos. Chem. Phys.* 20, 11869-11892. <https://doi.org/10.5194/acp-20-11869-2020>
- Boyer, M., Aliaga, D., Pernov, J.B., Angot, H., Quéléver, L.L.J., Dada, L., Heutte, B., Dall'Osto, M., Beddows, D.C.S., Brasseur, Z., Beck, I., Bucci, S., Duetsch, M., Stohl, A., Laurila, T., Asmi, E., Massling, A., Thomas, D.C., Nøjgaard, J.K., Chan, T., Sharma, S., Tunved, P., Krejci, R., Hansson, H.C., Bianchi, F., Lehtipalo, K., Wiedensohler, A., Weinhold, K., Kulmala, M., Petäjä, T., Sipilä, M., Schmale, J., Jokinen, T., 2023. A full year of aerosol size distribution data from the central Arctic under an extreme positive Arctic Oscillation: insights from the Multidisciplinary drifting Observatory for the Study of Arctic Climate (MOSAIC) expedition. *Atmos. Chem. Phys.* 23, 389-415. <https://doi.org/10.5194/acp-23-389-2023>
- Bozem, H., Hoor, P., Kunkel, D., Köllner, F., Schneider, J., Herber, A., Schulz, H., Leaitch, W.R., Aliabadi, A.A., Willis, M.D., Burkart, J., Abbatt, J.P.D., 2019. Characterization of transport regimes and the polar dome during Arctic spring and summer using in situ aircraft measurements. *Atmos. Chem. Phys.* 19, 15049-15071. <https://doi.org/10.5194/acp-19-15049-2019>
- Chételat, J., McKinney, M.A., Amyot, M., Dastoor, A., Douglas, T.A., Heimbürger-Boavida, L.-E., Kirk, J., Kahilainen, K.K., Outridge, P.M., Pelletier, N., Skov, H., St. Pierre, K., Vuorenmaa, J., Wang, F., 2022. Climate change and mercury in the Arctic: Abiotic interactions. *Science of The Total Environment* 824, 153715. <https://doi.org/https://doi.org/10.1016/j.scitotenv.2022.153715>
- Dastoor, A., Angot, H., Bieser, J., Christensen, J.H., Douglas, T.A., Heimbürger-Boavida, L.-E., Jiskra, M., Mason, R.P., McLagan, D.S., Obrist, D., Outridge, P.M., Petrova, M.V., Ryjkov, A., St. Pierre, K.A., Schartup, A.T., Soerensen, A.L., Toyota, K., Travnikov, O., Wilson, S.J., Zdanowicz, C., 2022a. Arctic mercury cycling. *Nature Reviews Earth & Environment* 3, 270-286. <https://doi.org/10.1038/s43017-022-00269-w>
- Dastoor, A., Wilson, S.J., Travnikov, O., Ryjkov, A., Angot, H., Christensen, J.H., Steenhuisen, F., Muntean, M., 2022b. Arctic atmospheric mercury: Sources and changes. *Science of The Total Environment* 839, 156213. <https://doi.org/https://doi.org/10.1016/j.scitotenv.2022.156213>
- Dastoor, A.P., Durnford, D.A., 2014. Arctic Ocean: Is It a Sink or a Source of Atmospheric Mercury? *Environmental Science & Technology* 48, 1707-1717. <https://doi.org/10.1021/es404473e>
- Dastoor, A.P., Larocque, Y., 2004. Global circulation of atmospheric mercury: a modelling study. *Atmospheric Environment* 38, 147-161. <https://doi.org/https://doi.org/10.1016/j.atmosenv.2003.08.037>
- Deng, L., Fu, D., Deng, N., 2009. Photo-induced transformations of mercury(II) species in the presence of algae, *Chlorella vulgaris*. *Journal of Hazardous Materials* 164, 798-805. <https://doi.org/https://doi.org/10.1016/j.jhazmat.2008.08.087>
- DiMento, B.P., Mason, R.P., Brooks, S., Moore, C., 2019. The impact of sea ice on the air-sea exchange of mercury in the Arctic Ocean. *Deep Sea Research Part I: Oceanographic Research Papers* 144, 28-

38. <https://doi.org/https://doi.org/10.1016/j.dsr.2018.12.001>
- Douglas, T.A., Blum, J.D., 2019. Mercury Isotopes Reveal Atmospheric Gaseous Mercury Deposition Directly to the Arctic Coastal Snowpack. *Environmental Science & Technology Letters* 6, 235-242. <https://doi.org/10.1021/acs.estlett.9b00131>
- Durnford, D., Dastoor, A., 2011. The behavior of mercury in the cryosphere: A review of what we know from observations. *Journal of Geophysical Research: Atmospheres* 116. <https://doi.org/https://doi.org/10.1029/2010JD014809>
- Durnford, D., Dastoor, A., Figueras-Nieto, D., Ryjkov, A., 2010. Long range transport of mercury to the Arctic and across Canada. *Atmos. Chem. Phys.* 10, 6063-6086. <https://doi.org/10.5194/acp-10-6063-2010>
- Ferrari, C.P., Gauchard, P.-A., Aspomo, K., Dommergue, A., Magand, O., Bahlmann, E., Nagorski, S., Temme, C., Ebinghaus, R., Steffen, A., Banic, C., Berg, T., Planchon, F., Barbante, C., Cescon, P., Boutron, C.F., 2005. Snow-to-air exchanges of mercury in an Arctic seasonal snow pack in Ny-Ålesund, Svalbard. *Atmospheric Environment* 39, 7633-7645. <https://doi.org/https://doi.org/10.1016/j.atmosenv.2005.06.058>
- Fisher, J.A., Jacob, D.J., Soerensen, A.L., Amos, H.M., Steffen, A., Sunderland, E.M., 2012. Riverine source of Arctic Ocean mercury inferred from atmospheric observations. *Nature Geoscience* 5, 499-504. <https://doi.org/10.1038/ngeo1478>
- Fushiki, T., 2011. Estimation of prediction error by using K-fold cross-validation. *Statistics and Computing* 21, 137-146. <https://doi.org/10.1007/s11222-009-9153-8>
- Galf, M., Lizotte, M., Kieber, D.J., Randelhoff, A., Hussherr, R., Xue, L., Dinasquet, J., Babin, M., Rehm, E., Levasseur, M., 2021. DMS emissions from the Arctic marginal ice zone. *Elementa: Science of the Anthropocene* 9, 00113. <https://doi.org/10.1525/elementa.2020.00113>
- Gong, X., Kaulfus, A., Nair, U., Jaffe, D.A., 2017. Quantifying O₃ Impacts in Urban Areas Due to Wildfires Using a Generalized Additive Model. *Environmental Science & Technology* 51, 13216-13223. <https://doi.org/10.1021/acs.est.7b03130>
- Hirdman, D., Aspomo, K., Burkhart, J.F., Eckhardt, S., Sodemann, H., Stohl, A., 2009. Transport of mercury in the Arctic atmosphere: Evidence for a spring-time net sink and summer-time source. *Geophysical Research Letters* 36. <https://doi.org/https://doi.org/10.1029/2009GL038345>
- Iversen, T., 1984. ON THE ATMOSPHERIC TRANSPORT OF POLLUTION TO THE ARCTIC. *GEOPHYSICAL RESEARCH LETTERS* 11, 457-460. <https://doi.org/10.1029/GL011i005p00457>
- Jin, X., Cai, X., Huang, Q., Wang, X., Song, Y., Zhu, T., 2021. Atmospheric Boundary Layer—Free Troposphere Air Exchange in the North China Plain and its Impact on PM_{2.5} Pollution. *Journal of Geophysical Research: Atmospheres* 126, e2021JD034641. <https://doi.org/https://doi.org/10.1029/2021JD034641>
- Kamp, J., Skov, H., Jensen, B., Sørensen, L.L., 2018. Fluxes of gaseous elemental mercury (GEM) in the High Arctic during atmospheric mercury depletion events (AMDEs). *Atmos. Chem. Phys.* 18, 6923-6938. <https://doi.org/10.5194/acp-18-6923-2018>
- Kirk, J.L., St. Louis, V.L., Hintelmann, H., Lehnerr, I., Else, B., Poissant, L., 2008. Methylated Mercury Species in Marine Waters of the Canadian High and Sub Arctic. *Environmental Science & Technology* 42, 8367-8373. <https://doi.org/10.1021/es801635m>
- Klonecki, A., Hess, P., Emmons, L., Smith, L., Orlando, J., Blake, D., 2003. Seasonal changes in the transport of pollutants into the Arctic troposphere-model study. *Journal of Geophysical Research: Atmospheres* 108. <https://doi.org/https://doi.org/10.1029/2002JD002199>

- Lanzillotta, E., Ceccarini, C., Ferrara, R., Dini, F., Frontini, F.P., Banchetti, R., 2004. Importance of the biogenic organic matter in photo-formation of dissolved gaseous mercury in a culture of the marine diatom *Chaetoceros* sp. *Science of The Total Environment* 318, 211-221. [https://doi.org/https://doi.org/10.1016/S0048-9697\(03\)00400-5](https://doi.org/https://doi.org/10.1016/S0048-9697(03)00400-5)
- Law, K.S., Stohl, A., 2007. Arctic Air Pollution: Origins and Impacts. *Science* 315, 1537-1540. <https://doi.org/10.1126/science.1137695>
- MacSween, K., Stupple, G., Aas, W., Kyllönen, K., Pfaffhuber, K.A., Skov, H., Steffen, A., Berg, T., Mastromonaco, M.N., 2022. Updated trends for atmospheric mercury in the Arctic: 1995–2018. *Science of The Total Environment* 837, 155802. <https://doi.org/https://doi.org/10.1016/j.scitotenv.2022.155802>
- Mann, E.A., Mallory, M.L., Ziegler, S.E., Avery, T.S., Tordon, R., O'Driscoll, N.J., 2015. Photoreducible Mercury Loss from Arctic Snow Is Influenced by Temperature and Snow Age. *Environmental Science & Technology* 49, 12120-12126. <https://doi.org/10.1021/acs.est.5b01589>
- Moore, C.W., Obrist, D., Steffen, A., Staebler, R.M., Douglas, T.A., Richter, A., Nghiem, S.V., 2014. Convective forcing of mercury and ozone in the Arctic boundary layer induced by leads in sea ice. *Nature* 506, 81-84. <https://doi.org/10.1038/nature12924>
- Pabi, S., van Dijken, G.L., Arrigo, K.R., 2008. Primary production in the Arctic Ocean, 1998–2006. *Journal of Geophysical Research: Oceans* 113. <https://doi.org/https://doi.org/10.1029/2007JC004578>
- Park, K.-T., Lee, K., Kim, T.-W., Yoon, Y.J., Jang, E.-H., Jang, S., Lee, B.-Y., Hermansen, O., 2018. Atmospheric DMS in the Arctic Ocean and Its Relation to Phytoplankton Biomass. *Global Biogeochemical Cycles* 32, 351-359. <https://doi.org/https://doi.org/10.1002/2017GB005805>
- Payne, C.M., van Dijken, G.L., Arrigo, K.R., 2022. North-South Differences in Under-Ice Primary Production in the Chukchi Sea From 1988 to 2018. *Journal of Geophysical Research: Oceans* 127, e2022JC018431. <https://doi.org/https://doi.org/10.1029/2022JC018431>
- Prijith, S.S., Rao, P.V.N., Mohan, M., Sai, M.V.R.S., Ramana, M.V., 2018. Trends of absorption, scattering and total aerosol optical depths over India and surrounding oceanic regions from satellite observations: role of local production, transport and atmospheric dynamics. *Environmental Science and Pollution Research* 25, 18147-18160. <https://doi.org/10.1007/s11356-018-2032-0>
- Qi, D., Ouyang, Z., Chen, L., Wu, Y., Lei, R., Chen, B., Feely, R.A., Anderson, L.G., Zhong, W., Lin, H., Polukhin, A., Zhang, Y., Zhang, Y., Bi, H., Lin, X., Luo, Y., Zhuang, Y., He, J., Chen, J., Cai, W.-J., 2022. Climate change drives rapid decadal acidification in the Arctic Ocean from 1994 to 2020. *Science* 377, 1544-1550. <https://doi.org/10.1126/science.abo0383>
- Schroeder, W.H., Anlauf, K.G., Barrie, L.A., Lu, J.Y., Steffen, A., Schneeberger, D.R., Berg, T., 1998. Arctic springtime depletion of mercury. *Nature* 394, 331-332. <https://doi.org/10.1038/28530>
- Shaw, G.E., 1995. The Arctic Haze Phenomenon. *Bulletin of the American Meteorological Society* 76, 2403-2414. [https://doi.org/https://doi.org/10.1175/1520-0477\(1995\)076<2403:TAHP>2.0.CO;2](https://doi.org/https://doi.org/10.1175/1520-0477(1995)076<2403:TAHP>2.0.CO;2)
- Shi, J., Chen, Y., Xu, L., Hong, Y., Li, M., Fan, X., Yin, L., Chen, Y., Yang, C., Chen, G., Liu, T., Ji, X., Chen, J., 2022. Measurement report: Atmospheric mercury in a coastal city of Southeast China – inter-annual variations and influencing factors. *Atmos. Chem. Phys.* 22, 11187-11202. <https://doi.org/10.5194/acp-22-11187-2022>
- Simpson, W.R., von Glasow, R., Riedel, K., Anderson, P., Ariya, P., Bottenheim, J., Burrows, J., Carpenter, L.J., Frieß, U., Goodsite, M.E., Heard, D., Hutterli, M., Jacobi, H.W., Kaleschke, L., Neff, B., Plane, J., Platt, U., Richter, A., Roscoe, H., Sander, R., Shepson, P., Sodeau, J., Steffen, A., Wagner, T., Wolff, E., 2007. Halogens and their role in polar boundary-layer ozone depletion. *Atmos. Chem.*

- Phys. 7, 4375-4418. <https://doi.org/10.5194/acp-7-4375-2007>
- Skov, H., Hjorth, J., Nordstrøm, C., Jensen, B., Christoffersen, C., Bech Poulsen, M., Baldtzer Liisberg, J., Beddows, D., Dall'Osto, M., Christensen, J.H., 2020. Variability in gaseous elemental mercury at Villum Research Station, Station Nord, in North Greenland from 1999 to 2017. *Atmos. Chem. Phys.* 20, 13253-13265. <https://doi.org/10.5194/acp-20-13253-2020>
- Sommar, J., Andersson, M.E., Jacobi, H.W., 2010. Circumpolar measurements of speciated mercury, ozone and carbon monoxide in the boundary layer of the Arctic Ocean. *Atmos. Chem. Phys.* 10, 5031-5045. <https://doi.org/10.5194/acp-10-5031-2010>
- Steffen, A., Douglas, T., Amyot, M., Ariya, P., Aspmo, K., Berg, T., Bottenheim, J., Brooks, S., Cobbett, F., Dastoor, A., Dommergue, A., Ebinghaus, R., Ferrari, C., Gardfeldt, K., Goodsite, M.E., Lean, D., Poulain, A.J., Scherz, C., Skov, H., Sommar, J., Temme, C., 2008. A synthesis of atmospheric mercury depletion event chemistry in the atmosphere and snow. *Atmos. Chem. Phys.* 8, 1445-1482. <https://doi.org/10.5194/acp-8-1445-2008>
- Strong, C., Rigor, I.G., 2013. Arctic marginal ice zone trending wider in summer and narrower in winter. *Geophysical Research Letters* 40, 4864-4868. <https://doi.org/https://doi.org/10.1002/grl.50928>
- Wang, J., Xie, Z., Wang, F., Kang, H., 2017. Gaseous elemental mercury in the marine boundary layer and air-sea flux in the Southern Ocean in austral summer. *Science of The Total Environment* 603-604, 510-518. <https://doi.org/https://doi.org/10.1016/j.scitotenv.2017.06.120>
- Willis, M.D., Leaitch, W.R., Abbatt, J.P.D., 2018. Processes Controlling the Composition and Abundance of Arctic Aerosol. *Reviews of Geophysics* 56, 621-671. <https://doi.org/https://doi.org/10.1029/2018RG000602>
- Wu, Q., Tang, Y., Wang, S., Li, L., Deng, K., Tang, G., Liu, K., Ding, D., Zhang, H., 2020. Developing a statistical model to explain the observed decline of atmospheric mercury. *Atmospheric Environment* 243, 117868. <https://doi.org/https://doi.org/10.1016/j.atmosenv.2020.117868>
- Yadav, J., Kumar, A., Mohan, R., 2020. Dramatic decline of Arctic sea ice linked to global warming. *Natural Hazards* 103, 2617-2621. <https://doi.org/10.1007/s11069-020-04064-y>
- Yu, J., Xie, Z., Kang, H., Li, Z., Sun, C., Bian, L., Zhang, P., 2014. High variability of atmospheric mercury in the summertime boundary layer through the central Arctic Ocean. *Scientific Reports* 4, 6091. <https://doi.org/10.1038/srep06091>
- Zhang, L., Zhou, P., Zhong, H., Zhao, Y., Dai, L., Wang, Q.g., Xi, M., Lu, Y., Wang, Y., 2021. Quantifying the impacts of anthropogenic and natural perturbations on gaseous elemental mercury (GEM) at a suburban site in eastern China using generalized additive models. *Atmospheric Environment* 247, 118181. <https://doi.org/https://doi.org/10.1016/j.atmosenv.2020.118181>
- Zhao, Y., Xi, M., Zhang, Q., Dong, Z., Ma, M., Zhou, K., Xu, W., Xing, J., Zheng, B., Wen, Z., Liu, X., Nielsen, C.P., Liu, Y., Pan, Y., Zhang, L., 2022. Decline in bulk deposition of air pollutants in China lags behind reductions in emissions. *Nature Geoscience* 15, 190-195. <https://doi.org/10.1038/s41561-022-00899-1>
- Zheng, W., Chandan, P., Steffen, A., Stuppel, G., De Vera, J., Mitchell, C.P.J., Wania, F., Bergquist, B.A., 2021. Mercury stable isotopes reveal the sources and transformations of atmospheric Hg in the high Arctic. *Applied Geochemistry* 131, 105002. <https://doi.org/https://doi.org/10.1016/j.apgeochem.2021.105002>

Reviewer #3 (Remarks to the Author):

The authors have done a great job in addressing the reviewer comments. Still, there are two key statements in the abstract (L45-47) "... shows that oceanic evasion is not significant throughout the Arctic Ocean but mainly occurs in the Marginal Ice Zone (MIZ) due to the specific environmental conditions in that region." and in the conclusions (L367-369) "this study offers new insights by showing that oceanic evasion is not significant throughout the Arctic Ocean but mainly occurs in the MIZ" that are over-interpreting the results. How can the authors exclude that oceanic evasion of Hg⁰ can be significant from other parts of the Arctic Ocean during summer as well, i.e. that Hg⁰ evasion mainly occurs from the MIZ only? Elevated oceanic evasion has been suggested from the Barents Sea or from the Beaufort Sea alike (Sonke et al., 2018, PNAS, <https://doi.org/10.1073/pnas.1811957111>; Fig. 6a). I suggest to slightly reformulating these statements. I think we needed Hg⁰ concentrations or direct Hg⁰ flux measurements not only along the summertime leg of the Polarstern but from other parts of the Arctic Ocean as well to draw such a conclusion.

I'm unclear about the statement in the abstract in L44 "The dominant source (> 50%) is oceanic evasion". In L404 of the previous version of the manuscript the authors have stated that 36.7 % of the GEM variation cannot be explained by the model (see comment Q4 by Reviewer 3). Wouldn't it be correct to say that "> 50% of the explained GEM variability is caused by oceanic evasion"? Thank you for clarifying this.

Please add the dates and the respective position of the Polarstern to Figure 1a, in the same way as you did in Figure S9. It should immediately become clear in Fig. 1 whether the research vessel was driving/drifting clockwise or counterclockwise.

The manuscript reads very nicely!

Reviewer #4 (Remarks to the Author):

The manuscript has been substantially improved. Comments from all reviewers have been properly addressed. I think the manuscript is now acceptable for publication on Nature Communications.

Responses to Reviewers' Comments

Dear Reviewers:

Thank you for your comments on our manuscript entitled “The Marginal Ice Zone as a dominant source region of atmospheric mercury during Arctic summertime” (manuscript ID: NCOMMS-23-05574A) and for offering us an opportunity for revision. We believe the comments have further helped us improve the overall quality of the manuscript. Below are our detailed responses to reviewers' questions. Revised texts of the manuscript are marked in red in the responses to reviewers' comments.

Reviewers' Comments and response

Reviewer #3 (Remarks to the Author):

Q1. The authors have done a great job in addressing the reviewer comments. Still, there are two key statements in the abstract (L45-47) “... shows that oceanic evasion is not significant throughout the Arctic Ocean but mainly occurs in the Marginal Ice Zone (MIZ) due to the specific environmental conditions in that region.” and in the conclusions (L367-369) “this study offers new insights by showing that oceanic evasion is not significant throughout the Arctic Ocean but mainly occurs in the MIZ” that are over-interpreting the results. How can the authors exclude that oceanic evasion of Hg⁰ can be significant from other parts of the Arctic Ocean during summer as well, i.e. that Hg⁰ evasion mainly occurs from the MIZ only? Elevated oceanic evasion has been suggested from the Barents Sea or from the Beaufort Sea alike (Sonke et al., 2018, PNAS, <https://doi.org/10.1073/pnas.1811957111>; Fig. 6a). I suggest to slightly reformulating these statements. I think we needed Hg⁰ concentrations or direct Hg⁰ flux measurements not only along the summertime leg of the Polarstern but from other

parts of the Arctic Ocean as well to draw such a conclusion.

Re. Thanks a lot for your comments! Sonke et al. (2018) indeed suggested a net air-sea Hg exchange over the continental shelf areas due to heat transfer from continental regions and turbulence from sea ice rafting. In this study we focus on the central Arctic, i.e., a region that excludes continental shelf areas (see for example map attached, Figure 1). Our main point is that oceanic evasion is not prevalent throughout the ice-covered central Arctic but mostly happens in the MIZ. We have also revised the corresponding statements in the abstract and conclusion in the manuscript respectively, which were also shown below:

“**Line 43-46:** A potential source contribution function (PSCF) analysis further shows that oceanic evasion is not significant throughout the **ice-covered central** Arctic Ocean but mainly occurs in the Marginal Ice Zone (MIZ) due to the specific environmental conditions in that region.”

“**Line 367-369:** Furthermore, this study offers new insights by showing that oceanic evasion is not significant throughout the **central** Arctic Ocean but mainly occurs in the MIZ.”

In addition, the evasion flux calculated in the MIZ in this study is higher than the flux over continental shelf areas (include Sonke et al. 2018 in the list of references), and we also added this content to the manuscript (Line 327-328).

Figure 1. The general range of the central Arctic Ocean (CAO). This figure is from the Pauline Snoeijis-Leijonmalm et al., (2022) (Pauline Snoeijis-Leijonmalm et al. ,Unexpected fish and squid in the central Arctic deep scattering layer.Sci. Adv.8,eabj7536(2022).DOI:10.1126/sciadv.abj7536)

Q2. I’m unclear about the statement in the abstract in L44 “The dominant source (> 50%) is oceanic evasion”. In L404 of the previous version of the manuscript the authors have stated that 36.7 % of the GEM variation cannot be explained by the model (see comment Q4 by Reviewer 3). Wouldn’t it be correct to say that “> 50% of the explained GEM variability is caused by oceanic evasion”? Thank you for clarifying this.

Re. Thanks for your suggestion. We agree with you and have revised this statement as “more than 50% of the explained GEM variability is caused by oceanic evasion” based on your comment.

Q3. Please add the dates and the respective position of the Polarstern to Figure 1a, in the same way as you did in Figure S9. It should immediately become clear in Fig. 1

whether the research vessel was driving/drifting clockwise or counterclockwise.

Re. Thanks for your suggestion. We have added the respective position and the corresponding date information of the Polarstern to Figure 1a based on your comments, which was also displayed below:

Figure 2. (a). Spatial distribution of observed gaseous elemental mercury (GEM) concentrations in the Arctic Ocean during the summer legs (June–September) of the MOSAiC expedition. The color scale gives the GEM concentration. (b) The time series of GEM concentrations during the summer legs (June–September) of the MOSAiC expedition.

Reviewer #4 (Remarks to the Author):

The manuscript has been substantially improved. Comments from all reviewers have been properly addressed. I think the manuscript is now acceptable for publication on Nature Communications.

Re. Thanks for your positive comments!